# Explainability of Predictive Process Monitoring Results: Can You See My Data Issues?

**Ghada Elkhawaga** [1,2,*] , **Mervat Abu-Elkheir** [3] and **Manfred Reichert** [1]

1. Institute of Databases and Information Systems, Ulm University, 89081 Ulm, Germany
2. Faculty of Computers and Information, Mansoura University, Dakahlia 35516, Egypt
3. Faculty of Media Engineering and Technology, German University in Cairo, New Cairo 11511, Egypt
* Correspondence: ghada.el-khawaga@uni-ulm.de

**Abstract:** Predictive process monitoring (PPM) has been discussed as a use case of process mining for several years. PPM enables foreseeing the future of an ongoing business process by predicting, for example, relevant information on the way in which running processes terminate or on related process performance indicators. A large share of PPM approaches adopt Machine Learning (ML), taking advantage of the accuracy and precision of ML models. Consequently, PPM inherits the challenges of traditional ML approaches. One of these challenges concerns the need to gain user trust in the generated predictions. This issue is addressed by explainable artificial intelligence (XAI). However, in addition to ML characteristics, the choices made and the techniques applied in the context of PPM influence the resulting explanations. This necessitates the availability of a study on the effects of different choices made in the context of a PPM task on the explainability of the generated predictions. In order to address this gap, we systemically investigate the effects of different PPM settings on the data fed into an ML model and subsequently into the employed XAI method. We study how differences between the resulting explanations indicate several issues in the underlying data. Example of these issues include collinearity and high dimensionality of the input data. We construct a framework for performing a series of experiments to examine different choices of PPM dimensions (i.e., event logs, preprocessing configurations, and ML models), integrating XAI as a fundamental component. In addition to agreements, the experiments highlight several inconsistencies between data characteristics and important predictors used by the ML model on one hand, and explanations of predictions of the investigated ML model on the other.

**Keywords:** predictive process monitoring; machine learning eXplainability; XAI; outcome prediction; process mining; machine learning

## 1. Introduction

### 1.1. Problem Statement

Process mining is a scientific field at the intersection between Business Process Management (BPM) and data science. Process mining allows the ideal view of how the activities of a process should be performed (i.e., the *normative model* of the business process), to be contrasted with the way they are carried out (i.e., the *descriptive model*) [1]. Predictive process monitoring (PPM) [2], as a fundamental use case of process mining, supports stakeholders by making predictions about the future of a running business process instance. A business process corresponds to a sequence of executed events affected by decision points and involving a number of actors to achieve one or more goals of the business process [3]. Predictions related to running process instances enable stakeholders to make preventive decisions in case of expected undesired outcomes, expected delays, or resource congestion.

As stakeholder engagement is at the center of process mining tasks, performance and accuracy are not the only aspects that matter in the context of a PPM prediction task. When depending on an ML model for predicting the future of running business

process instances, it becomes necessary to persuade stakeholders about the validity of the reasoning mechanisms applied by the predictive model. In particular, explaining predictions to the various stakeholders fosters users' trust, engagement, and advocacy in PPM-employed mechanisms.

In order to provide explanations of predictions generated by an ML model, eXplainable Artificial Intelligence (XAI) [4] methods and mechanisms [5–11] can be put in place. The explanations are obtained either in parallel to the prediction process or afterwards. Moreover, they are expected to reflect how a predictive model is influenced by the different choices made along the ML pipeline. The latter includes data analysis and cleaning, data preprocessing, model selection, parameter configuration, and evaluation of model predictions. Moreover, PPM tasks employ specific mechanisms for aligning process mining artefacts to ML model requirements. Despite increased awareness of the importance of explainability as a supportive mechanism for predictions in the context of PPM, to the best of our knowledge, no study of the effects of different choices made in the context of a PPM workflow has been carried out. Through such a study, it may be possible to understand the consequences of each choice in terms of its outputs, these being inputs in another step of the PPM pipeline. Through such a study, it might be possible to know beforehand which patterns are expected to affect an ML model, and hence which should be reflected in an explanation.

*1.2. Contributions*

This work attempts to study the connection between the characteristics of PPM inputs and steps on one the hand, and explainability outcomes on the other. In detail, this paper presents:

- A study of the effects that the underlying choices made in the context of a PPM task have on the predicted outcomes of a running process instance. In particular, we investigate changes in preprocessed data as a result of applying different transformation and preprocessing configurations.
- A study of how the explanations generated by two different global XAI methods (i.e., two model-specific methods, in addition to permutation feature importance and SHAP) can reflect inconsistencies and sensitivities in the executed predictive models. We investigate how changes in data that result from the application of different preprocessing configurations can expose the sensitivities of a predictive model and how this exposure can be reflected through explanations. Global methods used in the context of this research generate explanations for the outcomes of two predictive models. The latter are executed over process instances from three real-life event logs preprocessed with two different preprocessing configurations.
- An open-access framework of various XAI methods built upon different PPM workflow settings.

The rest of this paper is organized as follows: Section 2 provides background information on basic topics needed for understanding this work. In Section 3, we highlight the basic research questions investigated in this study. In Sections 4 and 5, our experimental settings, experimental results, and observations are discussed. Section 6 highlights the lessons learned and answers the basic research questions. Related works are discussed in Section 7. Finally, we conclude the paper in Section 8 with a summary and prospective outlook.

## 2. Background

This section deals with basic concepts and background needed for understanding this work. Section 2.1 introduces predictive process monitoring, with a focus on outcome-oriented predictions, as our study focuses on this prediction task. Section 2.2 summarizes available XAI methods.

### 2.1. Predictive Process Monitoring

A process instance is the execution of certain activities allowed in the context of a given business process. Predictive Process Monitoring (PPM) is a process mining use case that supports decision makers with predictions of the future course of a running process instance. This goal is realised by building models to generate predictions such as the next activity to be carried out, time-related information (e.g., the remaining time until process completion), the outcome of the process instance, execution costs, and executing resource [1]. The generation of a specific prediction is referred to as a *PPM task* .

A PPM task takes an event log as its central input; the event log documents the execution history of the instances of a business process in terms of *traces*. In turn, each trace represents the execution events of a single process instance.

For example, in an event log documenting a loan application process, each case, i.e., a process instance, represents a specific application process. Table 1 provides an example of a loan application process event log. Each case is composed of mandatory components and optional ones. Note that we use the terms case and process instance interchangeably here to refer to the same concept; however, we tend to use the term *case* where the data view of a business process is our concern and the term *process instance* when we are concerned with the conceptual view of a business process.

**Table 1.** Example loan application event log.

| Case ID | Activity | Applicant ID | Timestamp | Resource | Requested Amount | Granted Amount | Monthly Cost |
|---------|----------|--------------|-----------|----------|------------------|----------------|--------------|
| A150 | Create Application | C1820 | 30 March 2018 10:07:22 | John Doe | 38,000 | 30,000 | 1281 |
| A150 | Validate Application | C1820 | 30 March 2018 14:12:29 | Ben Markus | 38,000 | 30,000 | 231 |
| A150 | Decide | C1820 | 12 April 2018 11:15:30 | Jill Adams | 38,000 | 30,000 | 342 |
| A150 | Close Application | C1820 | 23 April 2018 13:09:11 | John Doe | 38,000 | 30,000 | 1213 |
| A218 | Create Application | C9121 | 15 March 2018 08:10:05 | Ben Markus | 57,000 | 44,000 | 643 |
| A218 | Validate Application | C9121 | 5 April 2018 15:02:22 | Ben Markus | 57,000 | 44,000 | 342 |
| A562 | Close Application | C7238 | 30 Dec 2018 10:29:11 | John Doe | 20,000 | 17,000 | 500 |

Table Legend: Categorical attribute / Numerical attribute

**Trace, Event Log.** Let $\varepsilon$ be the set of all possible events that may take place during the execution of a business process, i.e., the event universe. A trace $\sigma = < e_1, e_2, e_3, \ldots, e_n >$, with $n \in N$ being the total number of events that occurred during the execution of a process instance. A trace contains at least one event. Let D be the universe of data attributes associated with each event; then, $d_{ij} \in D$, where $i$ is the event number and $j \in M$ is the number of data attributes associated with the event.

A case corresponds to $< (e_1, (d_{11}, d_{12}, \ldots, d_{1m})), (e_2, (d_{21}, d_{22}, \ldots, d_{2m})), (e_3, (d_{31}, d_{32}, \ldots, d_{3m})), \ldots, (e_n, (d_{n1}, d_{n2}, \ldots, d_{nm})) >$. An event with its associated attributes must not take place more than once in a case, i.e., no redundancy is allowed when storing events associated with the same case. An event log (L) corresponds to a set of cases, in which each event appears at most once in the entire event log [1].

As mandatory attributes, a case should contain [1]:

- Case ID: represents a unique identifier of the case in the whole event log.

- Event class: represents a step carried out in fulfilling the process instance. This step is the activity name. *Receive application*, *check documents*, *assess risks*, and *notify customer* are all examples of events carried out in a loan application process instance.

According to [12], the timestamp attribute is considered a mandatory piece of information associated with each event. It documents the time when an event took place. When events are associated with their timestamps, the ordering of events (and hence the entire event log) can be based on these timestamps [1].

A case may contain optional data items representing information about a single event. These items are called the data attributes or data payload. Sometimes they are denoted as *dynamic attributes*, as they have different values for each event of a particular case. Examples of data items include the name of the person responsible for checking the customer's legitimacy, i.e., resource, and the documents associated with the form-filling event. When considering Table 1, it can be observed that a single case may be represented by several records, each representing a single event along with its data payload.

In addition to dynamic attributes, there are static ones associated with each case. *Static attributes* have constant values for all events of a given case. They represent data about the case itself, and their values are not changed during case execution. The applicant ID is an example of a static attribute. Note that certain attributes have numerical values, while others are categorical. This distinction has an effect on the choice of how to process the data values represented by this attribute. Furthermore, in the exemplary event log from Table 1 we do not define the datatype for the timestamp attribute. In most PPM tasks, this attribute is used to compute sub-attributes, e.g., week, day, and hour.

### 2.1.1. PPM Workflow

According to the survey results reported in [2], every PPM task follows a workflow that comprises four steps, which are in turn organised along two stages, i.e., an offline and an online stage (cf. Figure 1). Each PPM task starts with an event log that is (pre-)processed offline. By offline (pre-)processing, we mean transforming the historical cases of a business process captured in an event log into a form suitable for training a predictive model. The outcome of this preprocessing is passed to the online stage, in which an incomplete process instance is processed online, i.e., at runtime, and supported by online predictions. The complete PPM workflow is shown in Figure 1.

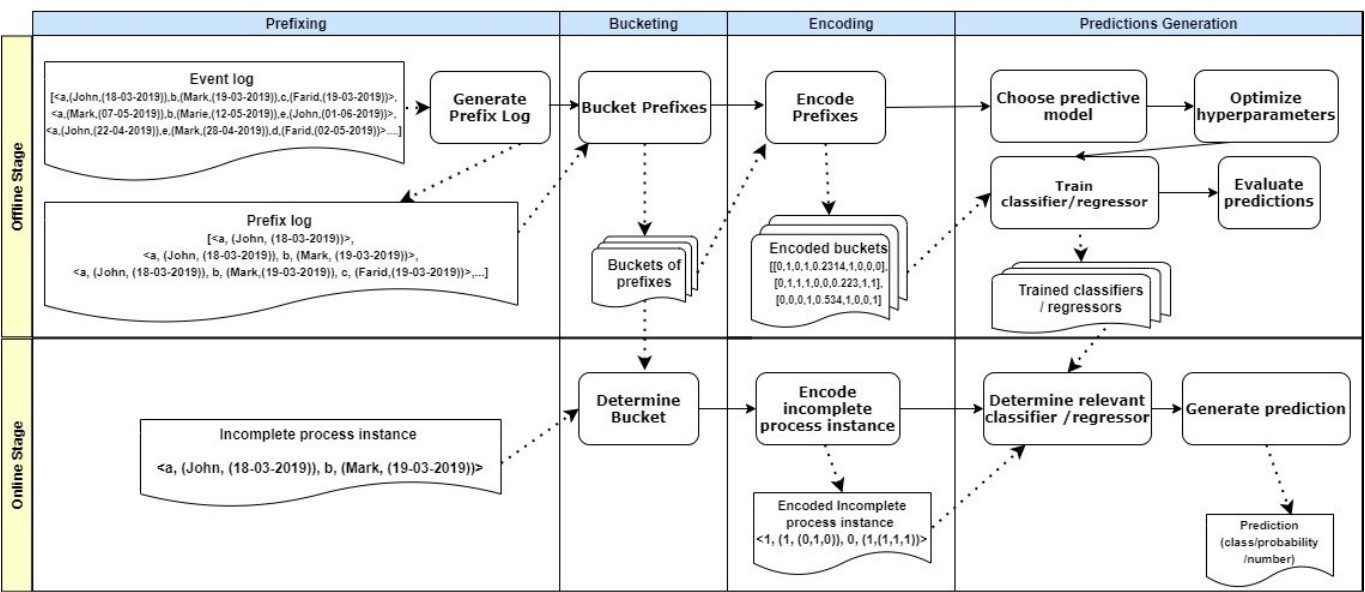

**Figure 1.** PPM Workflow.

1.  **PPM offline stage.** The input in this stage is an event log. Usually, the latter meets process mining requirements, i.e., it is organised in the form of cases. Each case has a

number of mandatory and optional attributes. Preprocessing steps are performed on the event log to transform its cases into a format compatible with the constraints and requirements of the prediction generation process.

(a) **Prefix log construction.** As stated by [2], the input of most PPM approaches relying on ML is a *prefix log* constructed from the input event log. Note that a predictive model is expected to predict information related to an incomplete process instance (i.e., a partial case). Consequently, the predictive model has to be trained on partial cases of historical process instances captured in the event log. Note that the prefixes generated from an event log may increase to the extent of slowing down the overall prediction process [2]. Therefore, prefixes can be generated by truncating a process instance up to a predefined number of events. As proposed by [13], truncating a process instance can be carried out up to the first k events of its case, or up to k events with a gap step (g) separating each two events, where k and g are user-defined. The latter prefixing approach is called *gap-based prefixing*.

**Prefix case, prefix log.** Consider a case $< e_1, (d_{11}, \ldots, d_{1m}), e_2,$ $(d_{21}, \ldots, e_n, (d_{n1}, \ldots, d_{nm}) >$, where $n \in N$ is the number of events and $m \in M$ is the number of event attributes. A prefix $P_k$ is generated from $\sigma$, where $P_1 = < e_1 >$, g is the gap or step size between an event and the following one in $P_k$, $1 \leq k \leq N$, and $1 \leq g \leq N - 1$. A prefix log is the set of all prefixes generated from all cases (or a defined set of cases) in the event log during preprocessing. A prefix log may include events with or without their associated data attributes.

After obtaining a prefix log from the given event log, the prefixes need to be further preprocessed in order to serve as appropriate inputs for the predictive model. Prefix preprocessing steps include *bucketing* and *encoding*. Available preprocessing configurations are limited in the literature, as reported by [2,12]. Figure 2 shows an overview of these techniques, which are introduced briefly through our illustration of the bucketing and encoding preprocessing steps.

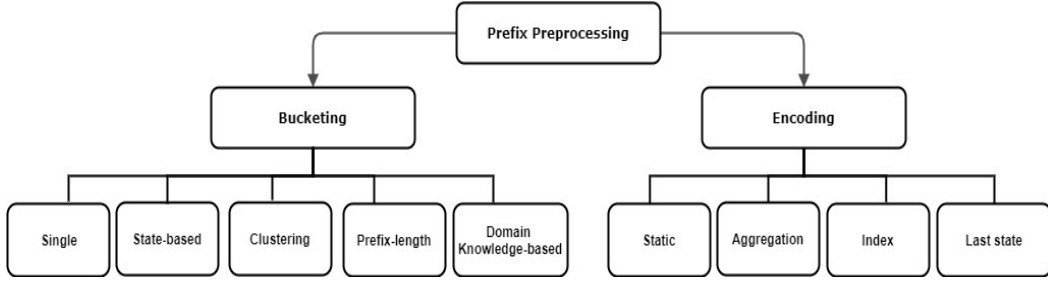

**Figure 2.** Preprocessing configurations (adopted from [2,12]).

(b) **Bucketing.** During prefix bucketing, the prefixes are grouped according to certain criteria (e.g., the number of events or the state reached during process execution). Prefixes in the same bucket are treated as a unit in the encoding and training steps of the offline stage. Moreover, in the online stage, these prefixes form a unit when examining an incomplete process instance in order to find its respective bucket, i.e., similar group of instances. Bucketing can be based on the commonly reached state of the grouped prefixes, on applying clustering techniques to group prefixes, on using domain knowledge or prefix length to bucket several prefixes, or simply on grouping all prefixes into one bucket, which is called the *single bucketing* technique.

In *single bucketing*, all prefixes generated from the cases of an event log are treated as a single bucket, whereas in *prefix length-based bucketing*, prefixes of the same length are bucketed together. Finally, for each bucket, a separate predictive model is created. Therefore, the choice of bucketing technique affects

the number of prefixes belonging to the same bucket as well as the number of predictive models to be built.

(c) **Encoding.** While bucketing techniques deal with how to group prefixes together, encoding techniques are concerned with how to format a single prefix in order to align it with the prediction process requirements. During prefix encoding, a prefix is transformed into a feature vector that serves as an input of the predictive model, either for training purposes or for making predictions. A predictive model can only receive numerical values. Therefore, all categorical attributes of a training event log or a training process instance need to first be transformed into a numerical form. The same applies to process instances with prefixes that serve as input for a predictive model. In the context of PPM-related encoding, a slight change to the regular ML encoding step is needed.

As reviewed in [2,12], there are four techniques used to encode prefixes. *Static encoding* is used to encode static attributes of a prefix event log, where numerical static features are used as-is and one-hot encoding is applied to categorical static attributes. In contrast, *aggregation, index-based* and *last state* encoding are examples of techniques used to encode dynamic attributes of prefix event logs. In *last-state encoding*, the last m states of a process instance are converted into a numerical form. This encoding technique is thus considered to be a *lossy technique*, as it leaves out important information about events that are executed before the selected last m states.

*Aggregation* encoding aggregates values of numerical dynamic attributes by using selected aggregation functions such as the sum, mean, or standard deviation of the values of the attribute for a single process instance. Furthermore, for categorical attributes all of the occurrences of the values of an attribute associated with a single process instance are either aggregated according to frequency-based or boolean-based counting (i.e., the attribute has a value or it does not). In *index-based* encoding, a process instance is represented by a single row, in which each value of each categorical attribute is the header of a column. Numerical values can thus be propagated as-is when using this encoding technique.

To illustrate both techniques, we refer back to Table 1 and take the process instance with *Case ID = A150*. Tables 2 and 3 provide two forms of this process instance, encoded using either *aggregation* or *index-based* encoding techniques, respectively. Note that we do not consider the *Timestamp* column in these examples, as this column is usually used to derive latent columns, e.g., hour, year, or day. However, the same encoding rules are applied to the latent columns according to the data type of these columns. Different encoding techniques yield different sizes of encoded prefixes with different types of included information. In turn, this diversity has been proven to affect both the accuracy of the predictions and the efficiency of the prediction process expressed in terms of execution times and needed resources [2].

**Table 2.** Feature vector created for case with id = A150 from the log in Table 1 using *aggregated* encoding.

| Case_Id | Applicant Id | Act_Create_app | Act_valid_app | Act_decide_app | Act_close_app | Res_John | Res_Benn | Res_Jill | Sum_monthly_cost | Requested_amount | Granted_amount |
|---------|--------------|----------------|---------------|----------------|---------------|----------|----------|----------|------------------|------------------|----------------|
| A150 | C1820 | 1 | 0 | 0 | 0 | 1 | 0 | 0 | 1281 | 38,000 | 30,000 |
| A150 | C1820 | 1 | 1 | 0 | 0 | 1 | 1 | 0 | 1512 | 38,000 | 30,000 |
| A150 | C1820 | 1 | 1 | 1 | 0 | 1 | 1 | 1 | 1854 | 38,000 | 30,000 |
| A150 | C1820 | 1 | 1 | 1 | 1 | 2 | 1 | 1 | 3067 | 38,000 | 30,000 |

**Table 3.** Feature vector created for case with id = A150 from the log in Table 1 using *index-based* encoding.

| Case_Id | Applicant Id | Act_1 | Act_2 | Act_3 | Act_4 | Res_1 | Res_2 | Res_3 | Monthly _cost_1 | Monthly _cost_2 | Monthly _cost_3 | Monthly _cost_4 | Requested _amount | Granted_amount |
|---------|--------------|-------|-------|-------|-------|-------|-------|-------|------------------|------------------|------------------|------------------|--------------------|----------------|
| A150 | C1820 | Create_app | valid_app | decide_app | close_app | John | Benn | Jill | 1281 | 231 | 342 | 1213 | 38,000 | 30,000 |

Note that the difference between the two examples is represented by the differences in how dynamic attributes are encoded. All encoding techniques applied to dynamic attributes are accompanied by *static encoding* used for encoding static attributes.

(d) **Predictive model construction and operation.** Depending on the PPM task, the respective predictive model is chosen. In this context, the prediction task type is not the only factor guiding the process of selecting the predictive model. Other relevant factors include the scalability of the predictive model when facing larger event logs, its simplicity, and the interpretability of the results. According to [2], Decision Trees (DT) are most often selected in current PPM research thanks to their simplicity. XGBoost represents another type of high-performing predictive model used in the context of PPM tasks [2].

The assignment of parameter values follows the model selection step. The values of model parameters are learned by the model during the training phase, whereas the values of hyperparameters are set prior to training the model. In the next step, the predictive model is trained based on encoded prefixes that represent completed process instances. Note that for each bucket, a dedicated predictive model needs to be trained, i.e., the number of predictive models depends on the chosen bucketing technique. After generating predictions for the training event log, the performance of a predictive model needs to be evaluated. Generally, the choice of the evaluation technique depends on the prediction task, e.g., classification tasks have different evaluation metrics than regression tasks.

2. **PPM online stage.** This stage starts with an incomplete process instance, i.e., a running process instance. Buckets formed in the offline stage are recalled to determine the suitable bucket for the running process instance. Finding the relevant bucket is based on the similarity between the running process instance and the prefixes in a bucket according to the criteria defined by the bucketing method.

For example, in the case of state bucketing, a running process instance is assigned to a bucket in which all process instances have the same state as the running process instance. Afterwards, the running process instance is encoded according to the encoding method chosen for the PPM task. The encoded form of the running process instance then qualifies as an input to the most relevant predictive model from those created in the offline stage. Finally, this stage is completed by the predictive model, which generates a prediction for the running process instance according to the pre-specified goal of the PPM task.

### 2.2. eXplainable Artificial Intelligence

Explainability, interpretability, and transparency are common terms with more or less the same meaning, referring to the problem of understanding and trusting the underlying mechanisms as well as the predictions of an ML model [14]. According to [15], an explanation is *"a human-interpretable description of the process by which a decision maker took a particular set of inputs and reached a particular conclusion"*. In the context of our research, the decision maker is an ML-based predictive model. Moreover, in this context explanation has a broader meaning, including both the reasoning process behind the generation of predictions and the factors that contribute to reaching a given prediction. For example, it is crucial to know which data characteristics influence a prediction (e.g., in cases of

imbalanced data or high correlations), as well as which features should be considered important to the predictive model.

Interpretability is *the degree to which a human can consistently predict the model's result* [4]. This human prediction and the mimicking of the model's reasoning is based on the mental model, a human form of explanation with respect to how the model reached its decisions. In [4,16], interpretability is considered equal to transparency. Transparency, however, can be regarded at three levels. The first level is *simulatability*, and refers to the human capability to simulate how a model reaches a prediction. The second level is *decomposability*, referring to the ability to understand a predictive model in terms of its elementary components, e.g., its inputs and hyperparameters. The last level, *algorithmic transparency*, denotes the understandability of the inner working of a prediction algorithm [4].

Model transparency itself can be an inherent characteristic or be achieved through explanation of a model. Transparent models are understandable on their own and satisfy one or all model transparency levels [16]. Linear models, decision trees, Bayesian models, rule-based learning, and General Additive Models (GAM) are all considered to be transparent and interpretable models.

However, the mental models of humans and our ability to mimic the reasoning process of a predictive model, and hence to trust and accept its predictions, depend on several qualities that determine a good explanation. The composition, content, and quality of an explanation can be influenced, characterised, and evaluated according to its constructing approach. XAI methods, in turn, can be differentiated along several factors, e.g., the way an explanation is generated or the granularity and scope within which an explanation can be generalised. Other factors may include the time at which an XAI method is applied, the way an explanation is presented, and the user group to which an explanation is presented.

Explanations can be generated in several ways, e.g., by providing *examples of process instances* which are similar in the values of their attributes but different in their prediction outcome. Another way is to *visualise* the intermediate representations and layers of a predictive model, with the aim of qualitatively determining what the has model learned [14]. *Explanation by feature relevance* is one type of XAI method, and includes assigning importance scores to those features that contribute to the prediction process of an ML model. Corresponding XAI methods include, e.g., SHAP [5] and Partial Dependence Plots (PDP) [6]. An explanation can be generated *locally* for one process instance or sample, or it may be applied *globally* to the reasoning process of a predictive model over the complete event log. Accumulated Local Effects (ALE) [9] is an example of a global XAI method, whereas LIME [8] is a local XAI method.

The presentation of an explanation depends on the way the explanation is generated, the characteristics of the end user (e.g., their level of expertise), the scope of the explanation, and the purpose of the explanation (e.g., to visualise effects of feature interactions on decisions of a given predictive model). Finally, Ref. [17] differentiates between *visual*, *verbal*, and *analytic* as three presentation forms of explanations.

The point in time at which an explanation is generated constitutes another relevant factor of an XAI method. When imposing explainability as an integral part of the predictive model, this is called *intrinsic explanation*, and the model is a white box. Finally, using an explanation method to understand the reasoning process of a model in terms of its outcomes is called *post hoc explanation* of a predictive model.

XAI methods are further categorised into *model-agnostic* and *model-specific*. Model-agnostic methods are able to explain any type of ML-based predictive model, whereas model-specific methods can only be used with specific models. For example, DeepLift [7] and LRP [10] provide explanations for predictive models based on neural networks. In [11], the authors discuss the advantages of model-agnostic methods, focusing on flexibility with respect to model choice, explanation form, and representation. In turn, Ref. [18] argues against the flexibility of model-agnostic methods and emphasizes that these approaches make assumptions about explained predictive models in order to maintain the expected flexibility.

## 3. Research Problems

The goal of our research is to study the effects of choices made through the PPM workflow on explanations as well as on how these explanations are able to reflect these choices. To this end, the research problem investigated here can be approached from two perspectives, the data side and the explanation side. It must be possible to understand the phenomena observed through explanations by understanding how the data transformation and preprocessing configurations change the original data. Furthermore, as illustrated in Section 2.2, interpretability and transparency implicitly accompany explainability and are inherent parts of it. On the way to achieving both, even partly, it should be possible for a stakeholder to trace a prediction back to its inputs. This should be done through a clear explanation that differs when the input data characteristics are different for the same ML predictive model and the same XAI method. Therefore, we have defined two Research Questions (RQ)s which capture this research problem on both sides.

**RQ1: What are the effects of applying different preprocessing configurations on the event log data in terms of changed characteristics and relations between features?** We need to study the expected changes resulting from the use of different preprocessing configurations on features characteristics (e.g., cardinality levels) and relations both between features (e.g., correlations) and with the target feature. By understanding how the applied bucketing and/or encoding techniques transform the characteristics of an event log, it becomes possible to choose suitable ML models while taking into account the sensitivities of these models to different data characteristics. Finally, by observing the changes caused by a specific preprocessing choice, we may be able to understand several observed phenomena through a single explanation.

**RQ2: How may explanations differ depending on the chosen preprocessing configurations?** After reaching an understanding of the changes occurring to the input data as a result of different preprocessing configurations, and considering the sensitivities of the used ML model(s), we need to understand how it is possible for an explanation to reflect different data characteristics. For example, what is the degree of similarity between the explanations generated by the same XAI method in explaining predictions of the same ML model with the same input data? In cases of dissimilarity, to what extent is this observation affected by the underlying data characteristics, e.g., collinearity between features? As another example, take linear models, which are transparent by nature [16], i.e., they are simulatable, decomposable, and algorithmically transparent. However, as confirmed by many studies [16,19], even transparent models (including linear ones) lose their simulatability and decomposability when facing high dimensionality or heavily engineered features. Therefore, preprocessing configurations which increase the dimensionality of the feature set may influence the explainability of certain predictive models. Moreover, self-explanatory mechanisms provided by the predictive model itself have to be differentiated in terms of stability over several runs of the model. Here, by stability we mean that the same feature set has the same importance scores after querying the predictive model for the important features several times. Henceforth, we refer to this analysis as a *stability check*.

## 4. Experiments

This section describes the choices of the experiments we performed in order to answer the two research questions introduced in Section 3. The PPM workflow (cf. Section 2.1.1) applies to all PPM tasks. In this study, we focus on predicting the outcome of a running process instance as a PPM task. However, the analyses conducted in this study are not PPM task-specific. For the basic workflow of experiments on PPM outcome prediction tasks (without an integrated explanations mechanism), we are inspired by the framework and findings presented in [2] and available at [20]. Note that, in order to take advantage of the reported performance of predictive models and preprocessing configurations from an explainability perspective, we have not changed any of the steps carried out in this framework.

Figure 3 shows the taxonomy of the implemented experiments organized under four dimensions to resemble an ML model creation pipeline, that is, aligned with the PPM offline workflow and incorporating an explainability-related dimension. Note that the data dimension has two blocks in the figure representing experiments performed before and after preprocessing. Categorising the experiments in terms of the designated taxonomy enables observation of the effect of making decisions in one dimension in isolation from the other dimensions where settings are kept unchanged. These dimensions are further investigated throughout this section.

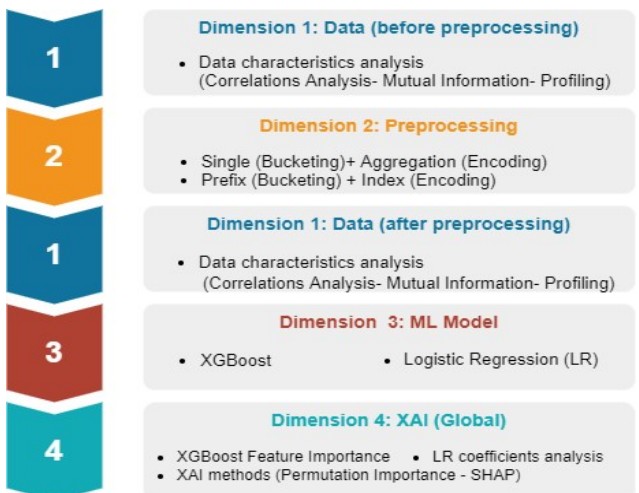

**Figure 3.** Experimental Taxonomy.

### 4.1. Experimental Setup

First, we describe the building blocks of our experiments, including the data, chosen preprocessing configurations, selected predictive models, and employed XAI algorithms. Categorising the experiments and the employed techniques (as shown in Figure 3) is accomplished with the aim of studying the impact of these choices on the resulting explanations. Observing the impact of modifications to different parameters in each dimension is the intended basis of this study. Note that all experiments were run using Python 3.6 and the scikit-learn library [21] on a 96-core a Intel(R) Xeon(R) Platinum 8268 @2.90 GHz with 768 GB of RAM. We applied all allowed combinations from each dimension using the taxonomy defined in Figure 3 in a dedicated experiment, while keeping choices relevant to other dimensions stationary. The code of the experiments is available through a Github repository (https://github.com/GhadaElkhawaga/PPM_XAI_Comparison, accessed on 10 August 2022) to enable open access for interested practitioners.

#### 4.1.1. Dimension 1: Data

The experiments were carried on three real-life event logs, which are publicly available from the 4TU Centre for Research Data [22]. One challenge in predictive process monitoring research is the limited availability of real-life event logs. Therefore, we depended on event logs available in Business Process Intelligence Challenges (BPIC) published by the 4TU Centre for Research Data [22]. We discarded logs that did not include a business process and those which did not include both static and dynamic attributes [2]. After filtering these, we obtained eight event logs, from which we selected three. The three basic event logs [22] used are as follows:

- **Sepsis:** This event log refers to the healthcare domain and reports cases of sepsis as a life threatening condition.
- **Traffic Fines:** This event log is a governmental one extracted from an Italian information system for managing road traffic fines.

- **BPIC2017:** This event log records instances of loan application processes in a Dutch financial institution.

These vary with respect to the referenced domain (healthcare, government, and banking), the number of cases (i.e., process instances), and the number of events per case. The chosen event logs further vary in the number of static and dynamic attributes, the number of categorical attributes, and as a result, the number of categorical levels available in each categorical attribute. We believe that these three logs represent a comprehensive basis for our experiments with respect to the five event logs that were left out. Choices in the data dimension are mainly impacted by the issues that need to be studied in **RQ1**.

The focus of our study is to analyze explainability of outcome predictions, i.e., to explain predictions of how a running process instance will terminate. Outcome predictions answer a specific question driven by the labels of the process instances, which are stored either in the training part of the event log or in its relevant testing event log. Outcome prediction is usually a binary classification problem [2]. In order to use the three chosen event logs in our experiments, we applied the same labelling functions defined in [2] to classify each process instance into one of two classes, i.e., a binary classification task. Regarding *Sepsis*, patients were classified according to

- whether a patient returned to the emergency room within 28 days of discharge in *Sepsis1*, and
- whether a patient was admitted to intensive care in *Sepsis2*.

Process instances in *Traffic_fines* are binary classified according to whether the fine was repaid in full or was sent for credit collection. Furthermore, instances from the *BPIC2017* event log are classified by whether the loan application was accepted or refused, resulting in *BPIC2017_Accepted* and *BPIC2017_Refused* event logs. As a result of applying these labelling functions, the number of considered event logs increased from three to five. A predictive model built for this classification task was then used to predict the labels we created in this step.

We chose event logs with different positive class ratios varying between <1% and 47% to study the effects of imbalanced event logs on explanations of different granularity. Table 4 shows basic statistics of the event logs used in our experiments. All event logs were cleaned and transformed according to the rules suggested by the framework presented in [2]. For example, traces were cut before the event used in the labelling step. Moreover, the event logs were enriched with derived attributes, e.g., the number of concurrently running cases at the time of executing the current event. Other examples of derived attributes include the position of an event in the case and the weekday, hour, and month, which were extracted based on the timestamp of each event. In order to avoid a dimensionality explosion after the encoding step, for each categorical attribute, category levels that appeared more than ten times were the only ones kept. Those not meeting this requirement were filtered out and replaced with the word "other" in the event log.

**Table 4.** Event logs statistics.

| Event log | #Traces | Short. Trace len. | Avg. Trace len. | Long. Trace len. | Max prfx len. | #Case Variants | %pos Class | #Event Class | # Static Col | # Dynamic Cols | #Cat Cols | #num Cols | #Cat Levels Static Cols | #Cat Levels Dynamic Cols |
|---|---|---|---|---|---|---|---|---|---|---|---|---|---|---|
| Sepsis1 | 776 | 5 | 14 | 185 | 20 | 703 | 0.0026 | 14 | 24 | 13 | 28 | 14 | 76 | 38 |
| Sepsis2 | 776 | 4 | 13 | 60 | 13 | 650 | 0.14 | 14 | 24 | 13 | 28 | 14 | 76 | 39 |
| Traffic_fines | 129615 | 2 | 4 | 20 | 10 | 185 | 0.455 | 10 | 4 | 14 | 13 | 11 | 54 | 173 |
| BPIC2017_Accepted | 31413 | 10 | 35 | 180 | 20 | 2087 | 0.41 | 26 | 3 | 20 | 12 | 13 | 6 | 682 |
| BPIC2017_Refused | 31413 | 10 | 35 | 180 | 20 | 2087 | 0.12 | 26 | 3 | 20 | 12 | 13 | 6 | 682 |

The analyses related to this dimension are intended to pinpoint the data characteristics and relations that may affect the patterns learned by a predictive model. Types of analysis in this dimension include:

- *Correlations analysis.* Correlation coefficient constitutes a measurement used to describe the degree to which two variables are linearly related [23]. It takes values between $-1$ and 1, with higher values indicating a stronger relationship. The sign

represents the direction of the relation. Correlation coefficient is computed to allow the normalised values of features to be investigated. For original event logs, we computed the correlations between categorical attributes before encoding them with Cramer's V coefficient [24]. The latter measures the correlation between two nominal variables and is based on Pearson's chi-square.

- *Mutual Information (MutInfo).* Correlations between features indicate the degree to which those features are dependent. However, correlations are not decisive with respect to the independence of features [23]. This means that if two variables are independent, their correlation coefficient equals zero. However, note that a correlation coefficient of zero does not indicate the independence of two variables [23]. We measured the MutInfo of features with respect to the label. MutInfo is the reduction of uncertainty in a variable after observing the dependent one [23]. This analysis is capable of capturing any kind of dependency, unlike the F-test, which captures only linear dependency [25]. Results take values between 0 and 1, with 0 meaning that the predicted label is independent of the feature and 1 meaning that both are totally dependent.

- *Profiling an event log.* For each event log, we generated a statistical profile called the *pandas profile* [26] both before and after preprocessing. Each pandas profile reports on statistical characteristics of each feature within the event log. Such characteristics include, for example, descriptive statistics of the feature, quantile statistics, missing values, most frequent values, and histograms.

Experiments along the data dimension were applied to the original event logs before starting the preprocessing step of the PPM workflow, i.e., before bucketing and encoding an event log. Afterwards, the same experiments were repeated on the transformed event logs. In this way, we analysed the effect of the increased dimensionality of an event log on the relationships between latent features that result from feature encoding. In addition, in the case of categorical attributes, experiments were performed in order to study which levels were highly correlated to other attributes.

### 4.1.2. Dimension 2: Preprocessing

As discussed in [2,12], the chosen bucketing technique might have an effect on the perceived performance of the predictive models. The more information available to the predictive model at the time the prediction is generated, the more accurate the prediction. Bucketing techniques can affect the information available to a predictive model if the bucketing prefixes are in the same execution state, have the same length, or if all prefixes from all traces are bucketed together. Here, we want to study whether the influence of a bucketing technique can be propagated and become clearer in explanations generated for a given prediction.

Moreover, the encoding technique equips the predictive model with different sets of features. As a result, its predictions are influenced by the characteristics of the encoding techniques. Examples of encoding technique characteristics include information loss or increased dimensionality [2]. The obligation of using either shorter traces or traces with gaps results from the latter characteristic.

In order to bucket traces of a chosen event log, we applied prefix-length bucketing with a gap of five events in addition to single bucketing. We believe that this gap size may enable a balanced situation to be reached in which only few prefixes are produced in case of event logs with longer traces, while enough prefixes remain available in case of event logs with shorter traces. Furthermore, the choice of a moderate gap size prevents a large number of events from being left out and avoids the risk of information loss, unlike in the case of larger gap sizes. Finally, this choice avoids overloading the experiments with large numbers of buckets when using a prefix size-dependent bucketing technique or single bucketing technique, unlike in the case of smaller gap sizes.

Moreover, we applied aggregation and index-based encoding techniques. Usually, both techniques are coupled with static encoding in order to transform static attributes

for use as input to the predictive model along with their dynamic counterparts. Two configurations of bucketing and encoding techniques were applied to satisfy both pre-processing tasks. These configurations were *single-aggregation* and *prefix-index*. As index encoding leads to a dimensionality explosion (because each categorical level of each feature is encoded as a separate column), we decided to apply this encoding only to the following event logs: *Sepsis* (i.e., the two derived event logs), *Traffic_fines*, and *BPIC2017_Refused*. Our decision to choose these two configurations, i.e., single-aggregation and prefix-index, was driven by several factors.

- Considering that certain encoding techniques turn subcategories of categorical attributes into separate features, we wanted to study how the relationship between the resulting new features is reflected though an explanation.
- We needed to study the ability of explanations in order to reflect differences in the reasoning process of a predictive model when all information is available in one input bucket. We aimed to compare the former situation against others when several predictive models are trained based on different groups of traces that differ in their lengths, and hence in the amount of information provided to a single predictive model.
- Our goal was to ensure the coherence of each scenario when a PPM pipeline is supported with explanations on top of it. By coherence, we mean that a scenario employs the best-performing techniques so as not to result in explanations which might be affected by limitations of the underlying choices. For example, we excluded the *last-state* encoding technique from our experiments, as its working mechanism ends with a scenario in which the predictive model is provided with limited information that is often insufficient for making accurate predictions [2].

  We chose *single-aggregation* and *prefix-index* because they have the least information-lossy techniques (i.e., index encoding) or the most comprehensive techniques to enable the input of various sizes of prefixes to the same predictive model (i.e., single bucketing). Finally, the results reported in [2,12] show that these configurations enable the building of predictive models with good performance. We applied single-aggregation to the five labelled event logs. After applying prefix-index (with a gap of five events) to the same event logs, we obtained thirteen event logs. At this point, we executed our experiments on eighteen event logs in total.

Tables 5 and 6 enumerate a few statistics about the event logs after applying *single-aggregation* and *prefix-index*, respectively.

**Table 5.** Statistics after applying *single-aggregation* to the event logs.

| Event Log | Training Bucket Size | Testing Bucket Size | # Features |
|---|---|---|---|
| Sepsis1 | 8974 | 2297 | 175 |
| Sepsis2 | 7222 | 1848 | 174 |
| Traffic_Fines | 362,094 | 88,530 | 254 |
| BPIC2017_Accepted | 494,892 | 124,815 | 722 |
| BPIC2017_Refused | 494,892 | 124,815 | 722 |

**Table 6.** Statistics after applying *prefix-index* to the event logs.

| Event Log | Prefix Length | Training Bucket Size | Testing Bucket Size | # Features |
|---|---|---|---|---|
| | 1 | 620 | 156 | 99 |
| Sepsis1 | 6 | 618 | 156 | 243 |
| | 11 | 531 | 140 | 425 |
| | 16 | 226 | 158 | 543 |

**Table 6.** *Cont.*

| Event Log | Prefix Length | Training Bucket Size | Testing Bucket Size | # Features |
|---|---|---|---|---|
| | 1 | 620 | 156 | 99 |
| Sepsis2 | 6 | 614 | 154 | 240 |
| | 11 | 468 | 124 | 408 |
| Traffic_Fines | 1 | 103,652 | 25,923 | 201 |
| | 6 | 8736 | 1965 | 901 |
| | 1 | 25,130 | 6283 | 120 |
| BPIC2017_Refused | 6 | 25,118 | 6283 | 1143 |
| | 11 | 24,952 | 6283 | 4104 |
| | 16 | 24,589 | 6261 | 7214 |

As discussed in Section 2.1.1, preprocessing configurations transform an event log into a format suitable for machine learning. From Table 5, we can observe the large bucket sizes of both training and test event logs preprocessed using *single* bucketing compared to their counterparts preprocessed using *prefix* bucketing, as presented in Table 6. Furthermore, *index* encoding is responsible for loading the event logs with a large number of features, as indicated in Table 6, compared to the number of features resulting from applying *aggregation* encoding (cf. Table 5). Event log preprocessed using *prefix-index* configuration comprise a lower number of prefixes having larger number of features with increasing prefix length. A crucial part of our study is to observe the effect of these transformations on data characteristics and generated explanations.

### 4.1.3. Dimension 3: ML Model

We are interested in explaining the predictions of process instance outcomes. This type of prediction task constitutes a binary classification task. Here, we employ two predictive models, XGBoost and Logistic regression (LR). Our target is to study the way explainability outcomes reflect the sensitivities of the predictive model being explained. Our choice of these models was based on two reasons:

- Both models provide mechanisms to highlight the most important features used in generating their predictions. In LR, the weights of the features may serve this purpose, whereas in XGBoost, a built-in capability can be used to retrieve feature sets and rank them by their importance to the model. Criteria used to rank features include *gain*, *weight*, and *cover*, according to the XGBoost API documentation [27]. Two complementary importance criteria are available through the scikit-learn implementation of XGBoost. These criteria are *total_gain* and *total_cover*.
- XGBoost is the best performing model according to the findings reported in [2], and is one of the best performing according to [12]. In order to build on the results and findings of these benchmarks with respect to different prediction tasks in predictive process monitoring, especially outcome predictions, we decided to adopt XGBoost.

Table 7 reports the performance of both models after training them in three iterations and using them to generate predictions for all the event logs. Note that both predictive models do not perform well on *Sepsis1*, although these results can be justified when considering the imbalance between class labels indicated in Table 4.

### 4.1.4. Dimension 4: XAI

Explainability experiments are divided into two sets, i.e., experiments depending on intrinsic explainability and experiments depending on post hoc XAI methods. *Intrinsic explainability* is achievable through querying mechanisms provided by the predictive model for retrieving features that influence prediction generation. We compared the explainability of predictive models to the results generated by each applied XAI method.

**Table 7.** AUC scores of predictive models categorized according to preprocessing configurations.

| Event Log | Preprocessing | XGboost | Logistic Regression |
|---|---|---|---|
| Sepsis1 | single_agg | 0.33427 | 0.57124 |
| | prefix_index (avg. AUC of len 1–16, gap = 5) | 0.4292 | 0.52599 |
| Sepsis2 | single_agg | 0.91374 | 0.88567 |
| | prefix_index (avg. AUC of len 1–11, gap = 5) | 0.81379 | 0.46253 |
| Traffic_fines | single_agg | 0.73918 | 0.7949 |
| | prefix_index (avg. AUC of len 1 & 6, gap = 5) | 0.6515 | 0.67804 |
| BPIC2017_Accepted | single_agg | 0.86429 | 0.8244 |
| BPIC2017_Refused | single_agg | 0.68328 | 0.70706 |
| | prefix_index (avg. AUC of len 1–16, gap = 5) | 0.7556 | 0.7677 |

To choose which XAI methods to apply, we considered criteria obtained from the explainability taxonomy available in [28]. Considering the explanations scale, in order to be able to explain the reasoning process of an ML model over the whole training event log, a global XAI method is needed. As intrinsic explainability is already satisfied by querying the used ML models themselves, post hoc XAI methods are necessary to validate the important features revealed by the models themselves. The chosen XAI methods have to be explaining by showing the relevance of features to the resulting predictions. Outcomes and explanations of the aforementioned methods are most comparable to the resulting set of important features when the ML model is queried. Furthermore, it is necessary to choose model-agnostic methods in order for the results to be applicable to predictions of both the LR and XGBoost models.

To this end, we used Permutation Feature Importance and SHAP (the global form). Moreover, we analysed LR coefficients and features important to XGBoost, as this provides a means to query them for important features. To check execution stability, we ran experiments with different settings and combinations of settings from the taxonomy in Figure 3 twice. Stability checks following this definition are expensive due to computational costs, which are affected by the number of choices used in the experiments for each dimension. As a result, it was not possible to run the full experimental settings combinations more than twice in the context of stability checks. However, in this study we are more concerned with a comprehensive experimental framework including several experimental dimensions.

The focus of stability checks in this dimension is on the stability of the explanations being generated by the model itself. Furthermore, we compared explanations of two global XAI methods over two runs of the experiments. In this study, we adopt a data-oriented focus and follow the data with analyses, starting from the event log and ending with the different explanations generated. A comparison of XAI methods with an in-depth analysis of their strengths along with their weaknesses constitutes a research gap that should be addressed; however, for this study these aspects are out of scope.

Figure 4 provides a summary of this section. For each of the selected event logs, we ran several experiments that follow the taxonomy depicted in Figure 3 and constitute steps in the following order.

1. Conduct data analysis of the event log features using three different techniques, namely, pandas profiling, correlation analysis, and mutual information analysis.
2. Preprocess the event log using *single-aggregation* and *prefix-index* configurations, which in turn produce several versions of the same event log.
3. For each event log resulting from the previous step,

   - Conduct data analysis using the same techniques as in step (1);
   - Train and build a separate predictive model using Logistic Regression and XGBoost;
   - Query each model for the important features it depends on (intrinsic explainability);

- Compare the most important features of each model over two runs (stability check);
- Compare the most important features of the two models with each other;
- Explain the predictions of the two models based on two XAI methods, i.e., Permutation Importance and SHAP (global explainability). We generated explanations twice and checked the similarity of the results in both runs (stability check).

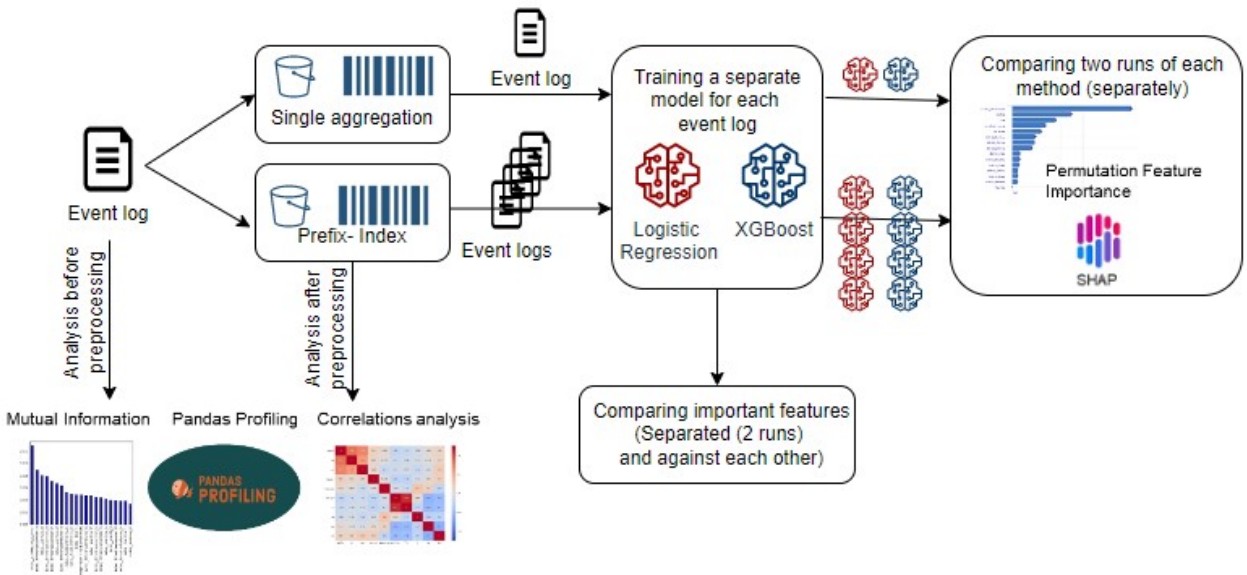

**Figure 4.** Experiments summary.

## 5. Results

It is important to view explanations in light of all contributing factors, including the characteristics of the input data, the effect of the preprocessing configurations, and the way in which certain predictive model characteristics affect its reasoning process. This section illustrates the observations we made during the experiments described in Section 4. We focus on the most remarkable observations and illustrate them by figures and tables. We categorise the obtained observations according to the respective perspective (either data and preprocessing or XAI). Finally, we sub-categorise the observations based on the goals of each of the two RQs defined in Section 3.

Due to lack of space, we cannot include the figures from all eighteen event logs we used in our experiments. Furthermore, some of the applied analysis techniques resulted in files with large sizes on disk and large numbers of columns; in particular, pandas profiles and correlation analysis produced large HTML documents and plots with hundreds of columns. Nevertheless, we build our observations on the results from these profiles. As the inclusion of the aforementioned artefacts is not possible in the context of this paper, we uploaded all plots and several other artefacts to a Github repository (https://github.com/GhadaElkhawaga/PPM_XAI_Comparison, accessed on 10 August 2022). We encourage researchers and practitioners to check this repository in order to ensure the repeatability and extension of these experiments as well as to reproduce the analysis and results.

### 5.1. Data- and Preprocessing-Related Observations

In the data analysis experiments we performed *before preprocessing* the event logs, we made several observations. Moreover, in the data analysis experiments we performed *after preprocessing* the event logs, we could observe the effects of preprocessing configurations. Studying the data characteristics before and after preprocessing the logs revealed a number of interesting observations, which are described in detail below.

**Data characteristics after preprocessing:**

*Observation 1:* *Several features that have a constant value before encoding show non-constant values after preprocessing depending on the applied encoding technique.*

**Example 1.** *in the* BPIC2017_Refused *event log, the feature "EventOrigin" has one value labelled "other" before preprocessing, and has 20 such values after applying frequency aggregation encoding. Moreover, certain features remain constant after encoding, e.g., "case:ApplicationType" in the same event log after its encoding based on frequency aggregations.*

*Observation 2:* *Certain features show a high number of zeros (i.e., have sparse vectors) before and after encoding.*

**Example 2.** *in Traffic_fines, the "points" feature has a sparse vector of about 97% zeros before encoding. This percentage is the same for both encoded versions of the same event log. However, the percentage of zeros increases in almost all event logs after preprocessing, as a result of applying either one-hot encoding in index encoding or counting the frequencies of categorical levels in aggregation encoding. The percentage of zeros reaches 99% in event logs encoded with index encoding, e.g., Traffic_fines.*

Moreover, after applying aggregation encoding techniques, certain event logs show a relatively low percentage of zeros, e.g., *Sepsis2*, which has about 33% of zeros, whereas other event logs have a high percentage of zeros, from 44 to 99%, e.g., *BPIC_2017*. **This observation can be justified by two factors**:

- *The effect of the bucketing technique.* Aggregation encoding is combined with single bucketing, which buckets all prefixes in the same group. Having many prefixes of the same process instance is more likely to reduce the effect of feature value imbalances. Such imbalances can happen due to the presence of prefixes generated from process instances with longer trace lengths. Moreover, index encoding is combined with prefix bucketing, which reduces the number of process instances fed into each encoding technique. As a result, combining index encoding with prefix-based bucketing has the potential to magnify imbalances in feature values.
- *The difference in 'zero' indication in both encoding techniques.* In aggregation encoding, a zero means that the feature did not have any value in the encoded event. In turn, in index encoding, a zero indicates whether the feature has a value in the process instance. Note that after aggregation encoding, a process instance might be represented along many rows, whereas after index encoding, it is represented by exactly one row. As a result, a high number of zeros does not guarantee the absence of a feature value after preprocessing certain events using aggregation encoding. A high number of zeros after index encoding, however, denotes the absence of a value. In summary, a feature might be considered for a predictive model even though it has a large percentage of zeros.

*Observation 3:* *Categorical levels extracted from certain categorical attributes dominate in prefix-indexed versions of certain event logs. This dominance increases with the length of the prefixes.*

This observation holds in the dominance of categorical levels of the "credit_score" feature in the prefix-indexed version of *BPIC2017_Refused* logs (cf. Figure 5).

**Data relations after preprocessing**

*Observation 4:* *Mutual Information, i.e., the dependency between features and the label in event logs, has low values in most cases.*

An exception to this observation holds for the single-aggregated version of *BPIC2017* logs. As a remarkable observation, labels in these logs depend on the feature "min_event_nr",

which remains among the top five highly dependent features with the label. In addition, these single-aggregated logs have a high dependency between aggregated forms of the "timesincecasestart" feature, which is the most dependent feature on the label in the original form of these logs. The same observation holds in the single-aggregated version of *Sepsis2*, whereas "remainingtime" depends on the label with a coefficient of nearly 0.4, reflecting similar dependency in the original log with a lower coefficient.

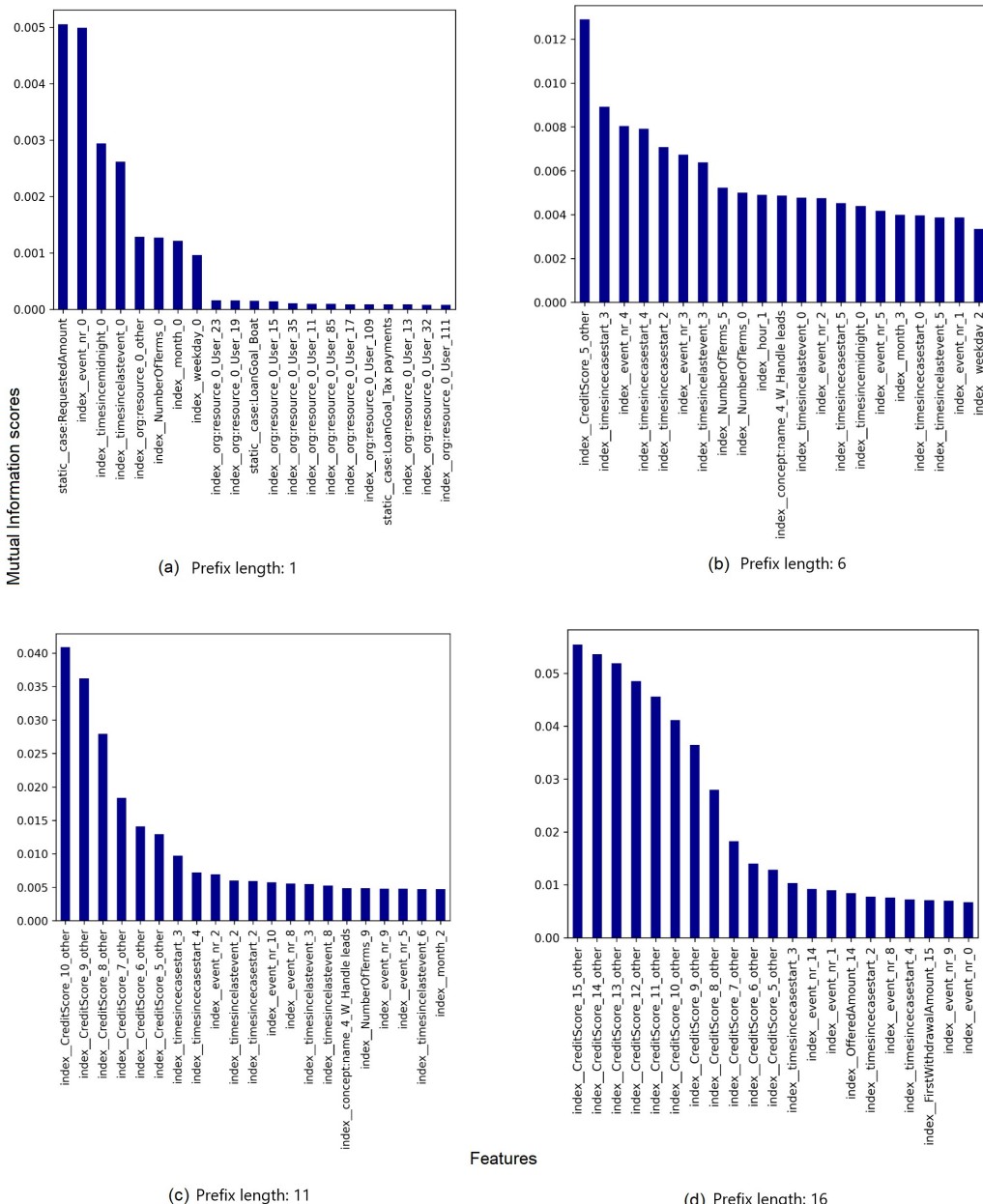

**Figure 5.** Mutual Information analysis of *BPIC2017_Refused* preprocessed with prefix index-configuration.

Studying correlations between features uncovers the following observations:

***Observation 5.1:*** *Two features are completely correlated across all event logs before and after applying the two preprocessing configurations, i.e., single aggregation and prefix index.*

To be more precise, these features are "hour" and "timesincemidnight". Note that these two features are artificial and their correlation is normal, as they are both extracted from

the same original DateTime column. This observation draws our attention to the possible effect that adding artificial columns might have on the prediction process to be conducted.

***Observation 5.2:*** *Certain domain-related features inherit correlations from the original logs and propagate this correlation in a strong form in the encoded versions of the same event log.*

**Example 3.** *"monthlycost", "numberofterms", "offeredamount", and "requestedamount" are strongly correlated in the two versions of the* BPIC2017 *event logs. After applying single aggregation preprocessing the correlations remain the same, followed by correlations between aggregated versions of computed attributes such as "timesincemidnight", "timesincelastevent", and "timesincecasestart". The same happens after applying prefix index preprocessing, with longer prefixes suffering from multicollinearity between category levels of the same attribute, e.g., "org:resource".*

**Example 4.** *This example is present in the complete correlations (i.e., those equal to 1) between categorical levels of certain features, e.g., "org:resource" and "dismissal", along with many other high correlations. These high correlations appear in event logs with longer prefixes in* Traffic_fines *more frequently than in event logs with shorter prefixes.*

**Example 5.** *Another example is manifested in the high collinearity between categorical attributes in original* Sepsis *logs, as shown in Figure 6 Such collinearity inherited as complete collinearity in those event logs obtained using either preprocessing configuration. However, the difference becomes obvious for numerical features in both preprocessing configurations. Partially, high correlations exist between aggregated forms of numerical features in single-aggregated event logs. Finally, only a few correlations exist between numerical features in shorter prefixes, increasing in number when prefixes become longer.*

***Observation 5.3:*** *High correlations emerge between features that are not correlated in the original log, while showing high correlations after applying both preprocessing configurations.*

**Example 6.** *There are correlations in* Traffic_fines *event logs after preprocessing in which certain correlated features contain constant values.*

We can summarize the former observations by indicating the effect of a preprocessing configuration in magnifying the absence of a feature value when it is not available in certain events of a process instance, rather than the whole trace. Another insight is that the applied preprocessing configurations generate many features which do not have a strong relation to the target while overloading the encoded event log with redundant features with strong collinearity.

The applied configurations load the encoded event logs with a large number of features derived from categorical features, despite the balance between categorical and numerical columns in three out of five event logs (*Traffic_fines, BPIC2017_Accepted, BPIC2017_Refused*). As a consequence, the high collinearity and dominance of these derived features with relevantly higher MutInfo values all result from the encoding techniques applied as part of the preprocessing configurations. The former observations are more obvious in event logs with longer prefixes, which demonstrates the key role of the choice of bucketing technique. With these observations, we reach the answer to **RQ1**.

*5.2. XAI-Related Observations*

5.2.1. Observations of Model-Specific Explanations

In the experiments described in Section 4, we apply LR and XGBoost. Both provide insights into the importance of features used in the prediction task through coefficients and feature importance in LR and XGBoost, respectively. As discussed later in this section, these methods do not always provide reliable results. However, the results can indicate which features influence the predictive model's decisions and their order regardless of the importance scores of these features.

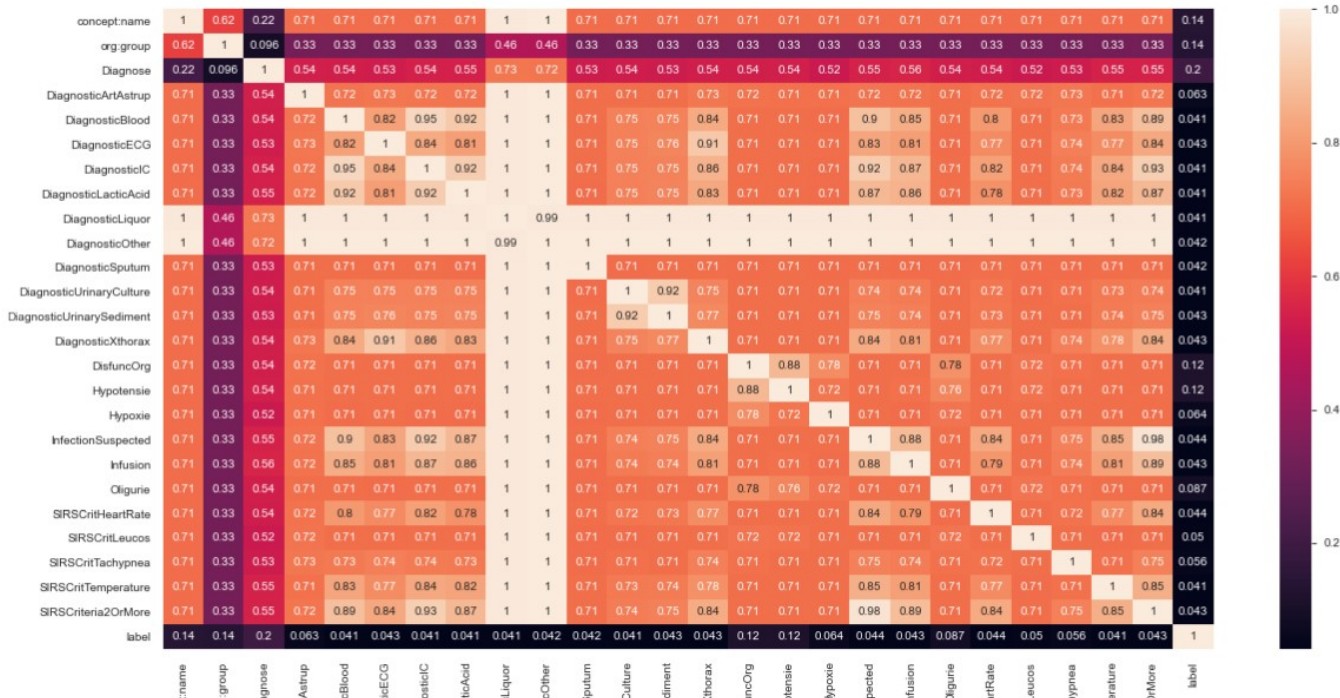

**Figure 6.** Collinearity between categorical attributes in *Sepsis2*.

Interpreting coefficients or weights of features in the context of LR, for example, is not a straight forward process the way it is in linear regression. LR employs a log function, and hence outputs the probability of belonging to a certain class [23]. Therefore, LR weights do not linearly influence the prediction. Instead, a change of a feature by one unit results in a change in the log odds ratio by the value of the corresponding weight of this feature [4]. Log odds means the logistic function of the probability of an event occurrence divided by the probability of no event [4].

**Effect of the applied preprocessing configurations**

When running LR over the used event logs twice, the following observations were made.

*Observation 6: Event logs preprocessed with the single-aggregation configuration show a 100% match in features coefficient analysis with respect to the features set and importance scores. However, the event logs do not show any similarity between the most important features, as indicated after querying the model and the features with high dependency on the label, according to MutInfo analysis.*

The *Traffic_fines* event log represents an exception to the latter observation. Analysis of the results of this event log show a similarity between the first two features with the highest LR coefficients and the first two features with high dependency on the label, according to MutInfo analysis. High collinearity between these two features is observed with respect to correlations analysis, as they constitute aggregations of "timesincelastevent" feature.

*Observation 7: Event logs preprocessed with the prefix-index configuration show similarities in feature coefficient analysis in terms of the features set with increasing prefix length. However, the importance scores remain different across several execution runs.*

Moreover, the similarity between the most important features according to the model and the set of features with high dependency on the label increases as the prefix length

increases. An exception to both observations with respect to prefix-indexed event logs is provided by the *Traffic_fines* event log. Here, the feature sets are not similar either when comparing them across the two runs of LR model coefficients analysis or when comparing them to the indicated features with high MutInfo values. As indicated in Table 4, *Traffic_fines* provides only shorter prefixes. Therefore, dissimilarity between feature sets across execution runs on this event log could be a result of its short prefixes. As a consequence, it is not possible to observe a change in longer traces, unlike in other event logs.

**XGBoost characteristics revealed when using the applied preprocessing configurations**

Referring back to the criteria provided by XGBoost to order features according to their importance, *gain* is considered to be the most relevant attribute in measuring the relative importance of each feature. We decided to rely on *gain* importance as provided by XGBoost, as it can be complementary to MutInfo, that is, the measurement representing the expected gain from a data analysis perspective. Comparing results from both a data analysis perspective and a model reasoning perspective, as well as observing how both sides (dis)agree, may lead to interesting insights which can highlight how a predictive model learns from the underlying data and reflects any anticipated characteristics. However, in our experiments we compare feature importance ranks based on all five criteria available in order to rank the importance of XGBoost features. These criteria are presented in Section 4.1.3. Note that in our observations there are two usages of the term *weight*, i.e., *weight*, indicating one of the five feature importance criteria analysed, and *the weight* indicating the importance score according to one of the five importance criteria. The following observations are made as a result of analysing features importance according to XGBoost-based models.

*Observation 8:* *XGBoost models show complete inconsistency across the two execution runs for all event logs preprocessed with the single-aggregation configuration.*

The two *Sepsis* event logs constitute an exception. They have the same feature set with different importance scores for four of the five criteria, whereas the feature set increasing the gain of the model is totally different across the two execution runs. However, in event logs preprocessed with the prefix-index configuration, models trained on prefix length-based event logs use the same feature sets with the same importance scores across the two execution runs for the five XGBoost criteria to rank important features. An exception to this consistency can be observed for prefix length-based event logs extracted from the *BPIC2017_Refused* event log.

*Observation 9:* *Multicollinearity, i.e., high correlations reaching complete correlations in certain cases, is an issue in event logs preprocessed with the single-aggregation configuration.*

There is an exception to this observation for the two event logs extracted from the *BPIC2017* log. This multicollinearity can be observed across the feature set of each criterion in the same execution run. Moreover, it can be observed in feature sets of the same criterion across the two execution runs.

*Observation 10:* *Feature sets which are important according to their gain differ from those with high dependency on the label, as indicated by the MutInfo analysis.*

This observation applies to all event logs, independently of whether they are preprocessed with single aggregation or prefix index configuration, across the two execution runs, except for the prefix length-based event logs extracted from the *BPIC2017_Refused* event log. For the latter event logs, the gain-based feature sets tend to match the MutInfo-based feature sets with increasing length of the prefixes in event logs.

***Observation 11:*** *In the event logs preprocessed with the prefix-index configuration, as prefix lengths increase the top important feature sets (according to their weight and cover criteria) become more similar across event logs.*

However, when comparing features across the same criterion (i.e., gain, weight, cover, etc.), we observe that, for similar features, the respective feature importance score decreases for longer prefixes. The *Traffic_fines* prefix length-based event logs provide an exception to this observation. This exception can be explained by the fact that these logs do not provide longer prefixes when applying a gap of five events; see the experiments settings in Section 4.1.

For example, Figures 7–9 compare LR coefficients with XGBoost features importance based on gain criterion. The comparison indicates increasing similarity concerning the important features subset with increasing prefix length.

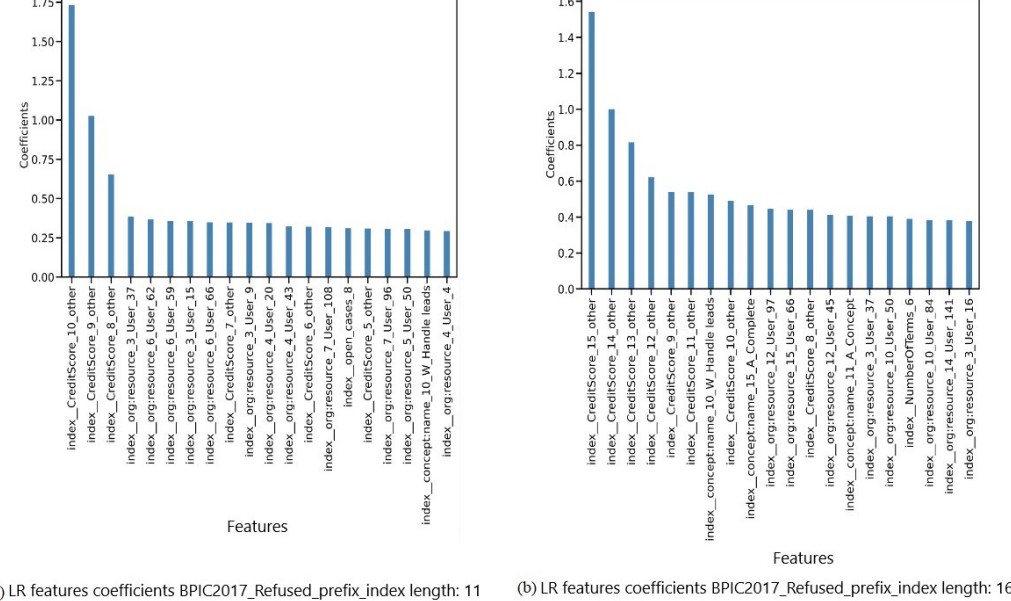

(a) LR features coefficients BPIC2017_Refused_prefix_index length: 11    (b) LR features coefficients BPIC2017_Refused_prefix_index length: 16

**Figure 7.** LR coefficients for different prefix lengths of *BPIC2017_Refused*.

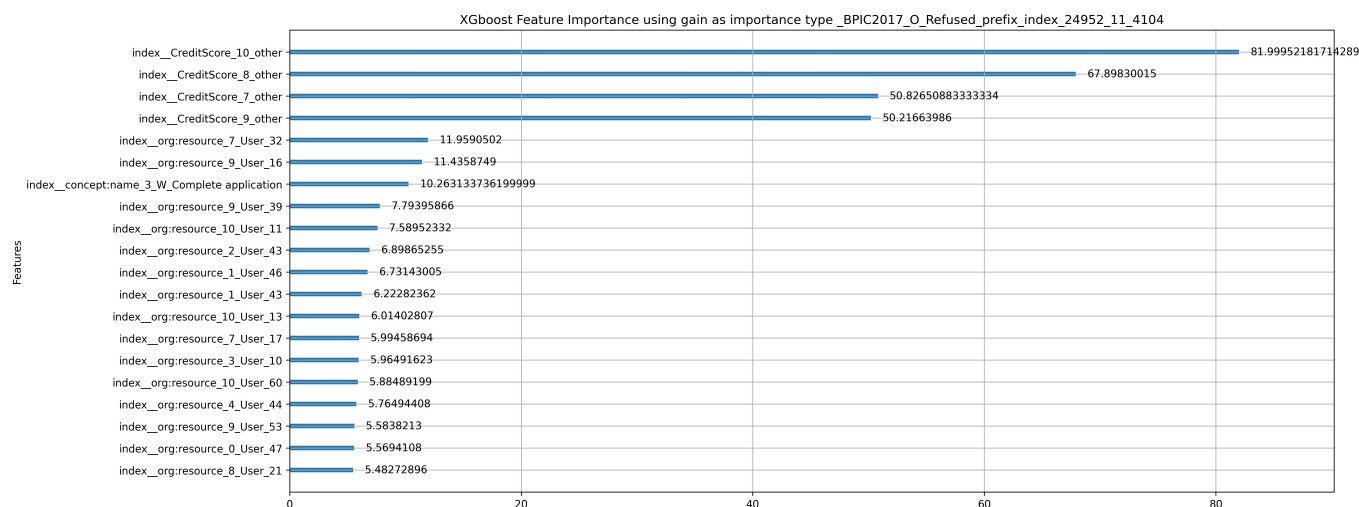

**Figure 8.** XGBoost features importance (gain) for prefix length (11) of *BPIC2017_Refused*.

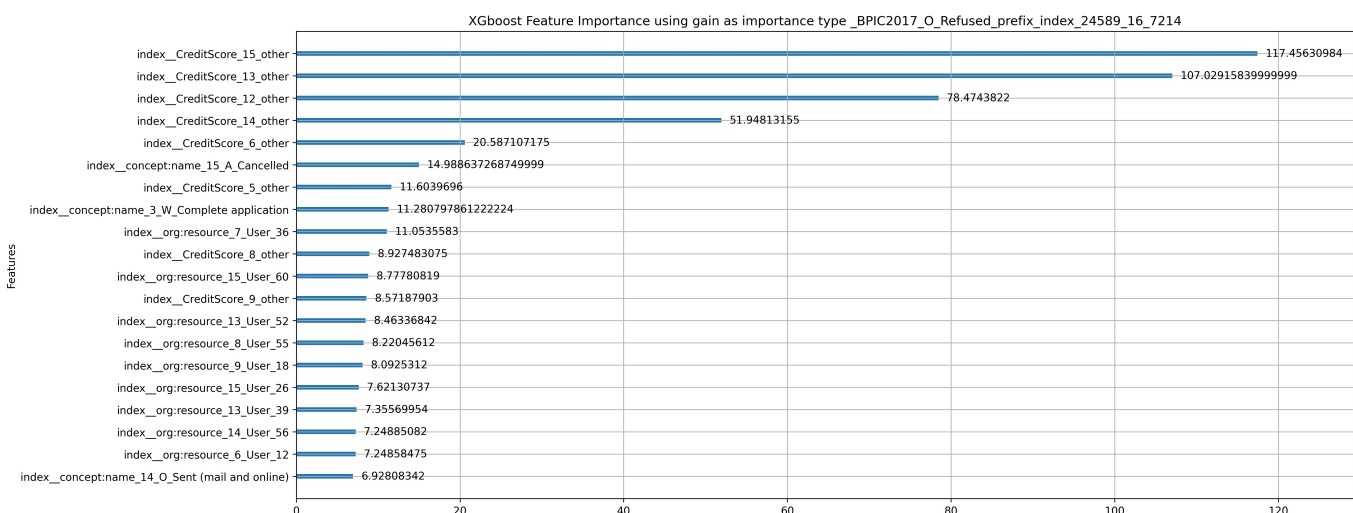

**Figure 9.** XGBoost features importance (gain) for prefix length (16) *BPIC2017_Refused*.

5.2.2. Observations of Global XAI Methods Results

In this subsection, an analysis of Permutation Feature Importance (PFI) and SHAP results is presented. For each XAI method, we search for indications of multicollinearity and (dis)conformance with important feature sets indicated by the employed predictive model.

**Permutation Feature Importance (PFI).**

The basic idea of PFI is to measure the average between the prediction error before and after permuting the values of a feature [4]. Feature values permutation or shuffling aims to estimate the increase in prediction error as an indicator of the feature importance. PFI was executed twice to query LR and XGBoost models trained over the event logs preprocessed with single-aggregation and prefix-index configuration. Each execution run included ten shuffling iterations. The mean importance of each feature was computed. In detail, PFI execution led to the following observations.

***Observation 12:*** *In single-aggregated event logs, the results of the two runs are consistent with respect to feature sets and the importance scores of these features. However, the resulting feature sets are affected by multicollinearity between the features. In prefix-indexed event logs, the two runs are consistent in all event logs.*

An exception in prefix-indexed event logs is present in the four logs derived from *BPIC2017_Refused*, with prefix lengths of [1, 6, 11, 16]. In the latter event logs, the dissimilarity between the feature sets across the two runs increases with increasing length of the prefixes. This observation can be attributed to the effect of the increased dimensionality in the event logs with longer prefixes. In prefix-indexed event logs, weights or scores of important features change with increasing prefix length.

***Observation 13:*** *(Dis)similarity between feature sets that results in the first run of PFI, on one hand, and LR coefficients or XGBoost feature sets on the other, is independent of the preprocessing configuration. In nearly all cases, dissimilarity can be observed.*

After comparing the results of the first PFI run with LR coefficients and XGBoost feature sets (based on the total gain criterion), complete dissimilarity can be observed between the PFI feature sets and LR coefficients regardless of the preprocessing configuration. Moreover, when compared to XGBoost feature sets, event logs showing no similarity in a single-aggregated form showed more similarity in certain features in prefix-indexed form with increasing length of the prefixes. Examples of this observation include the

*BPIC2017_Refused*, *Sepsis1*, and *Sepsis2* event logs. Figures 10 and 11 show PFI scores according to LR and XGBoost trained over *BPIC2017_Refused* of different prefix lengths.

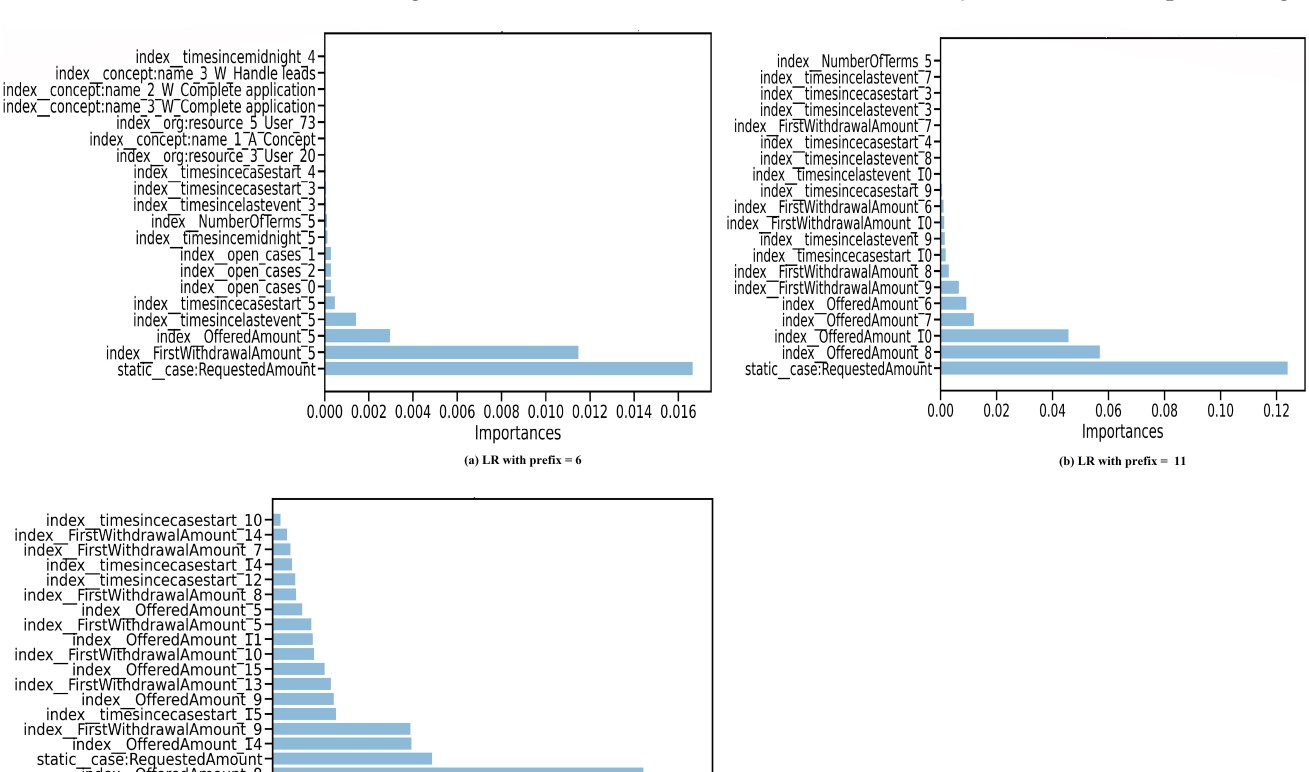

**Figure 10.** PFI according to LR trained over *BPIC2017_Refused* (preprocessed with prefix index configuration).

***Observation 14:*** *Multicollinearity between PFI feature sets of all single-aggregated event logs is at high levels, and increases with longer prefixes in prefix-indexed event logs.*

Multicollinearity implies features with very high or complete correlations amongst each other.

**SHapley Additive exPlanations (SHAP).**

SHAP is an explanation method that belongs to the class of feature additive attribution methods [5]. These methods use a linear explanation model to compute the contribution of each feature to a change in the prediction outcome with respect to a baseline prediction. Afterwards, a summation of the contributions of all features approximates the prediction of the original model. To maintain comparability of the global XAI methods used in our experiments, we constructed a SHAP explainer model on training event logs independently of another SHAP explainer model constructed from relevant testing event logs. The observations in this section are obtained based on the training SHAP explainer model.

***Observation 15:*** *While comparing the two execution runs, the results do not depend on the pre-processing configuration; rather, they differ depending on the respective predictive model for which explanations are generated.*

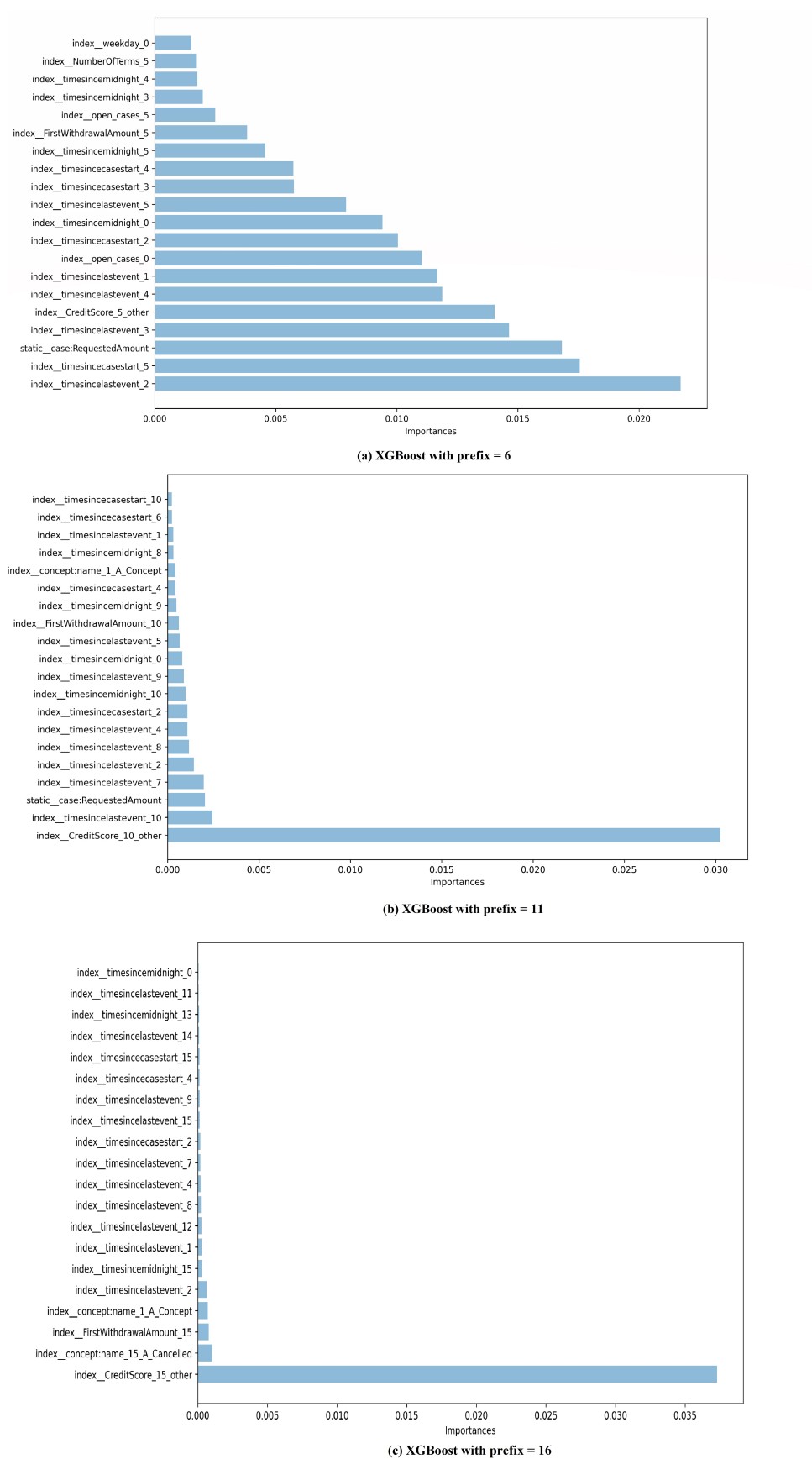

**Figure 11.** PFI according to XGBoost trained over *BPIC2017_Refused* (preprocessed with prefix index configuration).

In terms of explaining the predictions of the LR model, performing two executions of the SHAP method results neither in different feature sets nor in different ranks based on SHAP values, regardless of the preprocessing configuration used. In contrast, explaining the predictions of XGBoost model reveals that having the highest-contributing feature is the same across both execution runs. Furthermore, the rest of the feature set happens to be the same, with different respective ranks. An exception is present in the feature set of the two *Sepsis* event logs, where feature ranks are the same across both runs. These observations are aligned with the observations made regarding model-specific explanations and feature importance according to XGBoost-based models across two execution runs.

In order to study how SHAP results relate to feature importance as revealed by the predictive models, it becomes necessary to compare the features ranked according to their SHAP values with the important feature sets as indicated by the predictive models used in these experiments.

***Observation 16.1:*** *Important features according to their ranking based on their LR coefficients are unaligned with important features ranked by their SHAP values.*

This observation is valid for all considered event logs regardless of the preprocessing configuration.

***Observation 16.2:*** *When using the total gain as a criterion to rank important features according to XGBoost models, the former observation is not valid.*

In prefix-indexed event logs, the two compared feature sets are somehow similar. In contrast, in single-aggregated event logs, only in *BPIC2017_Accepted* and *Sepsis1* does there exist any similarity (though not a complete match) between the two compared feature sets.

***Observation 17:*** *Despite adjusting SHAP explainer parameters to be true to the model rather than the underlying relations between features, SHAP results reveal high multicollinearity between the most important features*

This observation especially holds for predictions based on the event logs preprocessed with the single-aggregation configuration. It is valid for the feature sets that are important to LR models, and is valid for the XGBoost models in the case of the *Sepsis1* event log as well. Multicollinearity in the underlying data affects LR, as its presence counteracts the underlying assumptions of LR. In contrast, multicollinearity is not supposed to affect XGBoost. This claim is justified by the fact that, in boosting, whenever collinearity exists between a subset of features the model chooses one feature to be the data splitting criterion. The entire importance score is assigned to the splitting feature, in contrast to the excluded correlated features, which are not considered important in this case [29]. However, in case of the *Sepsis1* event logs, the very high class imbalance (cf. Table 4) indicates that the model overfits patterns in training data.

Moreover, in event logs preprocessed with the prefix-index configuration, high correlations can be observed between features in the feature sets ranked highly according to LR with increasing prefix length. Multicollinearity in single-aggregated event logs is mainly inherited from high correlations present in original event logs. This multicollinearity increases with single aggregation preprocessing of an event log.

According to SHAP values, for certain event logs LR models depend on only one feature. The aforementioned logs are event logs with a prefix length of one, namely, those extracted from the event logs *BPIC2017_Refused*, *Sepsis1*, and *Sepsis2*. SHAP nullifies the contribution of features when they are not important to a predictive model. Therefore, for these event logs, only the important feature has a contribution value, while all other features receive zeros.

In Figure 12, features with values that are vertically allocated (around 0 in both sub-figures) are ignored by the predictive model. In contrast, features with values that are spread along the horizontal axis influence model predictions. This influence can be present through either lowering or increasing the values of model predictions. The color of the points corresponds to feature values in relevant process instances, independent of whether the value is high or low.

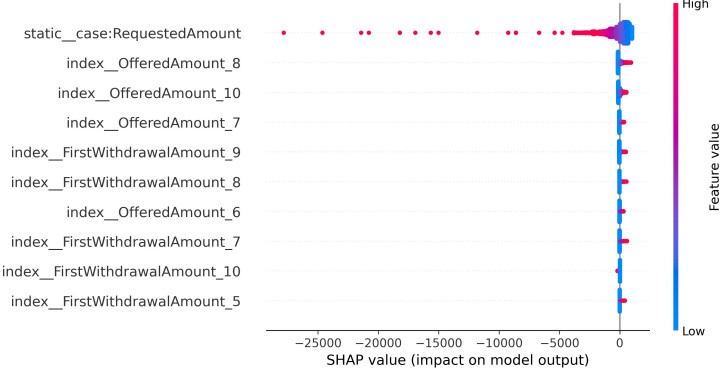

(**a**) SHAP values of BPIC2017_Refused with prefix = 11

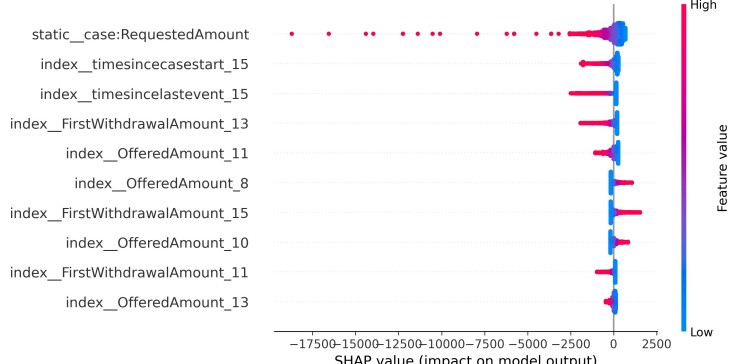

(**b**) SHAP values of BPIC2017_Refused with prefix = 16

**Figure 12.** Summary plots of SHAP values of features used by LR.

## 6. Discussion

Explaining ML-based predictions is a prerequisite for gaining user acceptance and trust in the predictions of respective models. It becomes necessary to consider explainability as a continuous process that needs to be integrated throughout the ML pipeline. A first step towards such integration would be to study the effects of different pipeline decisions on the resulting explanations. The main concern in our research is to study the ability of an explanation to reflect how a predictive model is affected by different settings in the ML pipeline. Another concern is to study how the inherent characteristics of a predictive model are confirmed and highlighted through an explanation. Our experimental results, described and analysed in Section 5, confirm the following conclusions:

- Encoding techniques used in the preprocessing phase of PPM have a major impact on both the ability of the predictive model to be decisive as to what the important features are and the selection of certain XAI methods to explain the model's predictions. Both studied encoding techniques load the event log with a large number of derived features. However, the situation is worse in index-based encoding, as the number of resulting features increases proportionally with the number of dynamic attributes, especially the number of categorical levels of a dynamic categorical attribute. In aggregation-based encoding, the number of resulting features increases in proportion with the number of dynamic attributes.

Increased collinearity in the underlying data constitutes another problem, resulting from encoding techniques in varying degrees. The effect of collinearity is observed in index-based preprocessed event logs, and is not totally absent in aggregation-based event logs. This collinearity is reflected in explanations of predictions with respect to process instances from prefix-indexed event logs as the length of a prefix increases.

- The selected bucketing technique has an effect on global XAI methods, especially when accuracy depends on the sufficiency of the process instances to be analysed. PFI is affected by the number of process instances as the average of errors in prediction is calculated over the number of event log process instances.

- Our experiments show the sensitivity of LR to collinearity in several situations. This conclusion can be made, for example, when comparing features ranked highly based on SHAP and PFI to high-importance features based on LR coefficients. In contrast, there is a degree of similarity when comparing the former important feature sets to XGBoost important features sets, especially for explanations of predictions from prefix-index preprocessed event logs. However, when querying the XGBoost model for the set of important features twice, the resulting sets do not match. Such a mismatch indicates inconsistency as a result of the collinearity between features. Note that both predictive models are affected by collinearity. However, in LR the effect is magnified, and prevails in all comparisons in which LR coefficients take part. In most cases, it is observed that dimensionality and collinearity prevent both LR and XGBoost from relying on features that have a dependency relationship with the label.

This discussion emphasizes the importance of considering the explanation of an event log as an accumulated effort, starting from features analysis and selection stage, to training a predictive model, and finally to an explanation model. Despite having XAI methods true to the model, explanations have the power to highlight how underlying data characteristics affect and are reflected in the model's reasoning process. The experiments executed in the context of this study are open to improvements and enhancements. For example, more real-life event logs are available; experimenting on more event logs with more varied characteristics may enable different observations to be obtained after the preprocessing step. Furthermore, despite using the most high-performing preprocessing configurations for our experiments, we believe that there is room for unexpected conclusions if the remaining set of preprocessing configurations are used.

## 7. Related Work

Over the last decade, a plethora of research has emerged that addresses PPM as an improvement and enhancement use case of process mining. Different techniques have been proposed and applied to predict different information about running process instances. These techniques are illustrated and compared in [30,31]. In [30], the authors categorise approaches based on prediction type, input type, algorithms used, and tool availability. They categorised the studied approaches into numerical, categorical, and next activity prediction-based approaches. A contribution of the study in [31] is the consideration of the process-awareness perspective when categorising runtime PPM approaches.

As an example of techniques that make use of the inherent transparency of the underlying predictive models, [32] uses Dynamic Bayesian Networks (DBN) to analyse the role of contextual attributes in predicting the next event of a running process instance. The former proposal benefits from the transparency of a DBN model, as the conditional probability distributions can be extracted at each state and time slice. A further example is proposed in [33], where a process-aware PPM approach is taken to predicting the remaining time of a running process instance while providing a transparent model. The latter paper provides an analysis of process performance using *flow analysis* techniques. The authors of [34] propose the use of fuzzy neural networks to predict the possible outcomes of running process instances. Through this proposal in [34] the authors discuss the transparency of the predictive model, as it is possible to extract the relations between input features and generated predictions in terms of IF–THEN rules.

As a complementary part of our present research, in [28] we study how characteristics of different XAI methods can be reflected in the generated explanations as well as how explainability methods can be compared according to different criteria. Furthermore, in [28], we propose an empirical analysis framework to study the impact of different PPM-related settings and ML model-related choices on the characteristics and expressiveness of the generated explanations. These explanations are generated on both global and local scales, i.e., for the whole event log and for individual process instances, respectively. We believe that the research introduced in this paper, along with the framework available in [28], constitutes a comprehensive framework. This framework enables study of the potential of explanations to reflect characteristics of contributing techniques and underlying phenomena taking place through the complete PPM workflow.

The authors of [35] propose an approach that integrates Layer-wise Relevance Propagation (LRP) to explain the next activity predicted using an LSTM predictive model. This approach tends to propagate relevance scores backwards through the model in order to indicate which previous activities were crucial to obtaining the resulting prediction. Another approach to explaining LSTM decisions is presented in [36]. According to this approach, the total number of process instances in which a certain feature contributes to a prediction is identified at each timestamp for the whole event log. This identification is directed by SHAP values. In [36], the authors use the same approach to provide local explanations for running process instances.

Explanations may be used to leverage predictive model performance, as proposed by [37]. Using LIME as a post hoc explanation technique to explain predictions generated using Random Forest, [37] identify feature sets which contributed to producing wrong predictions. After identifying these feature sets, their values were randomised, provided that they did not contribute to generating right predictions for other process instances. The resulting randomised event log was then used to retrain the model again until its perceived accuracy improved.

## 8. Conclusions

In this research, a framework is implemented to study the effects of several choices made in the context of a PPM task on the relationships between data before and after preprocessing. Furthermore, we analyze explanations of predictions made for business process instances. These explanations are generated using either self-explanatory predictive models or post hoc methods. We study the ability of these methods to reflect underlying characteristics of the input data, and hence to improve the transparency of the prediction process, at least from the input perspective.

Using two preprocessing configurations with two predictive models and two post hoc XAI methods may limit the generalisability of the observations and conclusions made in our experiments. However, we argue that our conclusions can provide useful insights about how certain preprocessing configurations lead to certain data characteristics that can expose sensitivities of certain predictive models, and hence become clear through the generated explanations. Furthermore, despite the limited number of techniques we experimented on, the same techniques have been reported in several studies to lead to the best performance results.

Our study reveals inconsistencies between data characteristics, the way an ML model uses these data, and the way such usage is reflected in the resulting prediction explanations. Our study highlights situations in which data problems do not affect the accuracy of predictions, only the usefulness and consistency of explanations. Therefore, explainability should be seamlessly integrated into PPM workflow stages as an inherent task, not as a follow-up effort. We believe our findings can help stakeholders to make more informed decisions about which techniques to apply through the PPM workflow in light of the analysis provided here.

**Author Contributions:** G.E. drafted and revised the manuscript. G.E. was responsible for conceptualizing this work. G.E. was responsible for the formal analysis, methodology, and writing the original

draft. M.R. was responsible for reviewing and editing this article. M.A.-E and M.R. validated and supervised this work. M.R. was responsible for funding acquisition. All authors read and agree to the published version of the manuscript.

**Funding:** This study was carried out through funding provided as part of the cognitive computing in socio-technical systems program granted to the last author, as the supervisor of the first author's PhD candidacy.

**Institutional Review Board Statement:** Not applicable.

**Informed Consent Statement:** Not applicable.

**Data Availability Statement:** Datasets used in the context of our experiments are available at 4TU Centre for Research Data (https://data.4tu.nl/Eindhoven_University_of_Technology, accessed on 10 August 2022), and the code for the experiments carried out in the context of this work is available at https://anonymous.4open.science/r/PPM_XAI_Comparison-1E8D/README.md, accessed on 10 August 2022.

**Conflicts of Interest:** The authors declare no conflict of interests.

## Abbreviations

The following abbreviations are used in this manuscript:

| | |
|---|---|
| ALE | Accumulated Local Effects |
| BPM | Business Process Management |
| GAM | General Additive Models |
| MutInfo | Mutual Information |
| ML | Machine Learning |
| PDP | Partial Dependence Plot |
| PFI | Permutation Feature Importance |
| PPM | Predictive Process Monitoring |
| RQ | Research Question |
| SHAP | SHapley Additive exPlanations |
| XAI | eXplainable Artificial Intelligence |
| XGBoost | eXtreme Gradient BOOsting |

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
