# Peer review of "Explainability of Predictive Process Monitoring Results: Can You See My Data Issues?"

_applsci, doi:10.3390/app12168192_

Round 1
Reviewer 1 Report
The paper is an exploration of various issues that can arise when performing predictive process monitoring. Specifically, the authors focus on effects (such as introduction of new correlations between features) related to data preprocessing (in the first phase of converting log into training set) and the post-hoc effects of data manipulation in early stages when performing XAI analysis.
I believe that the paper is a nice exploration based on 3 real-world logs and certainly deserves attention in our community (the reviewer publishes in process mining). The paper is reasonably readable and well-organized, yet it suffers from opaqueness, which I will relate to below in the list of weaknesses. I think that once Sections 3-5 are adjusted according to my review the paper can be accepted for publication. But for now, I recommend major revisions.
Strengths:
1. Exploring the effects of various design choices and decisions made when executing a PPM pipeline is an interesting topic.
2. The paper is well-organized – I like the separation into sections as provided by the authors. The structure is very clear and easy to follow.
3. The PPM pipeline in Section 2 is well-explained.
4. The results presented in Section 5 are interesting although they could be performed in a more (statistically speaking) scientific way and be less based on only visual cues.
Weaknesses:
1. Section 1: The paper is a bit opaque in places like the last paragraph of the introduction section. I think this part should be removed – it creates the feeling that the paper is not self-contained. I would mention citation [12] only in related work, but not after the paper organization has been presented.
2. Section 3: The formulation of the research questions should be made a bit more “formal”. What are the variables that you examine? What is the treatment you are testing? Who is the mediator? I am expecting a bit more “design of experiment” language. After reading RQ1 and RQ2 I remained puzzled as to what they mean.
For example:
Let X be data preprocessing (which can take different values, e.g., bucketing type).
Let Y be the outcome of interest (feature correlations, or XAI results).
Let Z be a mediator (not sure what that would be in your case).
Let W be a confounder.
Then a research question can be stated as: We would like to examine the direct effect X->Y and the indirect effect X->Z->Y.
Please investigate statistical literature (e.g., design of experiments (DoE), causal inference) to improve formulation.
I feel that the experiment could be better designed (in the sense of statistical DoE).
Related issue: the language used after each RQ is confusing. Consider adding the recommendations from 2, but also make the statements in a clearer fashion. Please make the discussion less vague.
3. Section 4: Better explain your design choices in Section 4 – for data: why these datasets? For measures, why MI? Why not others? It is unclear whether the experiments consider a representative sample PPM preprocessing methods and XAI methods. Why did you specifically select the XAI methods that you did? This needs to be better rationalized.
In general, Section 4 (all its 4 dimensions) require an overhaul that would depend on how you will re-formalize RQ1 and RQ2. You should clearly state what are the values that your controlled variables can receive. Same for uncontrolled variables and confounders. Currently, Section 4 is a bit all-over-the-place with a mix of bullet points and text, that does not concisely explain your own design choices.
4. Section 5: The experiment is a bit exploratory. It does not attempt to measure effects in a statistical manner (e.g., using bucketing reduces outcome by W%), but rather it comes as a list of observations which may be interpreted as cherry-picking or intuition that the authors gathered from the results. Yes, there are tables that compare figures, but was the difference significant? How did you measure the effect? Is it legitimate to compare methods in such way?
Statements like Observation 1, seem a bit arbitrary. Is this a generalizable finding? This seems to be very data specific. Please keep your findings statistically sound, show a hypothesis test that you ran for example, to test for correlations. Otherwise, it appears to be ad-hoc. What can we deduce from Obs 2? I find these statements of little value. Of course, some features will be sparse.
Bottom line: I would re-do (or extend) Section 5 after sections 3 and 4 are modified to be more precise, because all observations in Section 5 appear to be very exploratory and visual (rather than solid statistical tests).
5. What about inter-case feature engineering? Currently, you are only considering intra-case features, but recent work on inter-case analysis has shown the importance of those features. However, it is unclear what is the impact of inter-case log preprocessing. Please address this issue in the next version of the paper.
References:
Arik Senderovich, Chiara Di Francescomarino, Chiara Ghidini, Kerwin Jorbina, Fabrizio Maria Maggi:
Intra and Inter-case Features in Predictive Process Monitoring: A Tale of Two Dimensions. BPM 2017: 306-323
Arik Senderovich, Chiara Di Francescomarino, Fabrizio Maria Maggi:
From knowledge-driven to data-driven inter-case feature encoding in predictive process monitoring. Inf. Syst. 84: 255-264 (2019)
Minor issues:
1. Some of the plots show pixels (e.g., Table1 should be converted into LaTeX imho). Figure 1 is clear but again pixelized – please fix.
2. Line 118 – should be x \neq y, right?
3. Table 2 should be moved to top of the page. In general, tables and figures shouldn’t interrupt the flow and better present at the top.
Author Response
Please see the attachement

Reviewer 2 Report
COMMENTS.
1. The problem to be covered in this article is of interest. It provides information to be used as a basis, to be able to apply the prediction task considering the results obtained in this article.
The structure of the article is adequate and correct.
Points to highlight. The experimentation is very complete, the configuration to carry out the experimentation is fully detailed. This allows the results to be replicable.
--- Results.
1) An evaluation of the trained prediction models is not done, using the error metrics. For example, R2, RMSE, MAPE. Etc. Through these metrics it is possible to know how much it affects the processing of the data in the prediction models.
2) The type of cleaning that was applied to the event logs is missing in the text, did this cleaning change the flow of some cases? Were tasks eliminated? Or was any information discarded? Describe and clarify these doubts in the document.
3) If you chose to use another type of representation or coding of the event log, would the methodology be the same? Would it work?
4) The prediction task being evaluated needs to be described a little more. Is the next task being predicted?
Author Response
please see the attachement

Round 2
Reviewer 1 Report
The authors have addressed my comments in an adequate manner. I would recommend a one last round of grammar and fixing figures that look odd (as authors have promised to do upon acceptance).